# Identifying genetic variants associated with chromatin looping and genome function

**Sourya Bhattacharyya** [1] ✉ **& Ferhat Ay** [1,2] ✉

Here we present a comprehensive HiChIP dataset on naïve CD4 T cells (nCD4) from 30 donors and identify QTLs that associate with genotype-dependent and/or allele-specific variation of HiChIP contacts defining loops between active regulatory regions (iQTLs). We observe a substantial overlap between iQTLs and previously defined eQTLs and histone QTLs, and an enrichment for fine-mapped QTLs and GWAS variants. Furthermore, we describe a distinct subset of nCD4 iQTLs, for which the significant variation of chromatin contacts in nCD4 are translated into significant eQTL trends in CD4 T cell memory subsets. Finally, we define connectivity-QTLs as iQTLs that are significantly associated with concordant genotype-dependent changes in chromatin contacts over a broad genomic region (e.g., GWAS SNP in the *RNASET2* locus). Our results demonstrate the importance of chromatin contacts as a complementary modality for QTL mapping and their power in identifying previously uncharacterized QTLs linked to cell-specific gene expression and connectivity.

Using genotype to predict phenotypic responses to perturbations, whether at the molecular, cellular or organismal level, is the ultimate challenge of personalized medicine. Genome-wide association studies (GWAS) link common genetic variants to measurable phenotypes and disease susceptibility[1]. The majority of these GWAS variants are, however, present in the noncoding DNA sequences and are inherited as dense haploblocks[2], hence identifying functional or causal GWAS SNPs becomes challenging[3]. Expression quantitative trait loci (eQTL) studies either in bulk cells or in single-cell populations have quantified the effect of noncoding *cis* variants on gene expression for different tissues and cell types[4–15]. However, high degree of linkage disequilibrium (LD) among the derived eQTLs (bulk or single-cell) or GWAS SNPs make identifying the putative causal variants difficult. High-throughput functional validation approaches[16–19] as well as statistical approaches[20–31] were developed to identify or prioritize putatively causal variants for different diseases and cell types. Another set of approaches overlap these eQTLs or GWAS SNPs with cell type-specific maps of regulatory elements to further annotate their relation to gene expression and disease risk[32–36].

In parallel, breakthroughs in capturing the 3D genome structure using various chromatin conformation capture (3C) technologies such as Hi-C[37,38] and its variants including Promoter Capture Hi-C[39], PLAC-seq/HiChIP[40,41] led to genome-wide maps of cell-type-specific chromatin interactions/loops across many cell types and conditions[42,43]. These studies demonstrated the importance of cell-type-specific physical proximity between regulatory elements (e.g., enhancers) and target gene promoters, and demonstrated how subtle changes in such proximity may lead to meaningful differences in gene expression[44]. These studies also provided a rationale to integrate chromatin interactions with GWAS and eQTL statistics for identifying putative causal variants. A number of reference studies[45,46] including our previous work[47] identified enhancer regions harboring disease-associated SNPs that act on their target genes through looping from over long genomic distances. Other efforts integrated chromatin marks or chromatin QTLs with information from 3D genome organization to annotate and prioritize disease-associated SNPs[4,19,48–55].

Published works profiling cell-specific 3D chromatin conformation[45,47,56,57] in primary cells did it on a limited number of donors ($n = 2–6$) making it difficult to identify de novo associations between genotype and chromatin contacts. Among these works, Watt et al.[57] on primary neutrophils mapping transcription factor binding QTLs for PU.1 utilized allele-specific bias of promoter capture Hi-C

[1]La Jolla Institute for Immunology, La Jolla, CA, USA. [2]Department of Pediatrics, University of California, San Diego, La Jolla, CA, USA. ✉e-mail: sourya@lji.org; ferhatay@lji.org

(PCHi-C) data but did that on 1D coverage (all PCHi-C reads) of the heterozygous SNPs rather than individual chromatin loops involving the SNP region as an anchor. Similarly, our previous work on 6 donors also considered allele-specific variation of 1D coverage from HiChIP for a set of examples[47]. Other studies that derive associations between genotype and 3D genome organization use Hi-C which did not provide sufficient resolution to identify SNPs associated with specific loops[44,58]. For example, using Hi-C data from 20 lymphoblastoid B-cell lines (LCLs), Gorkin et al.[58] identified a few hundred QTLs associated with contact counts while for larger scale features such as directionality index, insulation score and compartment values, they reported thousands of QTLs. Greenwald et al.[44] generated phased Hi-C data from induced pluripotent stem cells (iPSCs) and iPSC-derived cardiomyocytes of 7 donors to derive 114 haplotype-associated chromatin loops (HTALs) primarily driven by imprinting and/or CNVs. Two more recent studies aim at identifying loop/interaction/contact-associated SNPs from circulating immune cell populations using Hi-C data from T cells[59] or eQTL-capture Hi-C data from monocytes[60]. Overall, there is still a need for genome-wide profiling of high-resolution contacts between regulatory elements and combining both genotype-dependent and allele-specific information of chromatin contacts to define QTLs associated with looping.

Here we present a comprehensive H3K27ac HiChIP dataset on naïve CD4 T cells (nCD4) from 30 donors and provide a framework to identify QTLs associated with genotype-dependent and/or allele-specific variation of HiChIP contacts. We denote the set of derived QTLs as interaction QTL or iQTL. Although the term interaction QTL has been used in studying GTEx tissues and modeling the interaction between cell type abundance and genotype (ieQTLs)[61], here we use it to denote the QTLs (SNPs) associated with HiChIP interaction strength. We perform a comprehensive comparison between the derived iQTLs and the reference nCD4 eQTLs to show that the use of chromatin interactions retrieves eQTLs enriched for fine-mapped disease-risk GWAS variants. In addition, nearly one third of loops with iQTLs overlapped histone QTLs (hQTLs) for H3K27ac on their anchors with high concordance between effect sizes from both QTL modalities suggesting that the underlying mechanism for such iQTLs is the genotype-associated variation in histone modifications. A significant fraction of the nCD4 iQTLs that are not nCD4 eQTLs became eQTLs in CD4 T cell memory subsets (e.g., Th1, Th2, Th17 or Tregs) such that the significant variation of chromatin contacts in nCD4 are translated into significant eQTL trends associated in specific subsets. In addition, we identified a set of SNPs that are iQTLs but not hQTLs in nCD4 or eQTLs in any CD4 T cell subset while overlapping potential regulatory regions and GWAS SNPs. This suggests the importance of chromatin contacts as a complementary data modality to eQTL and other molecular QTL mapping studies. Finally, we also define connectivity-QTLs, namely SNPs that are significantly associated with a group of HiChIP contacts within a broad region or connecting two such regions that are non-overlapping. Overall, using variation of chromatin interactions across donors, we identify regulatory and putatively functional SNPs, and connect the genotype-dependent variation of gene expression, either in naïve CD4 T cells or other CD4 T cell subsets, with that of specific chromatin loops connecting regulatory elements.

## Results

### Interaction QTLs (iQTLs): a class of QTLs associated with 3D chromatin interactions

As previous works showed the existence and importance of genetic variants that have a statistically significant association with gene expression (expression QTLs[4], splicing QTLs[61]) or epigenomic features (chromatin accessibility QTLs[62], histone QTLs, methylation QTLs[63], tfQTLs[57]), here we set out to define interaction QTLs (iQTLs) that are associated with the intensity of a chromatin contact between two genomic regions (H3K27ac HiChIP contact counts in

CD3+CD4+CD45RA+CCR7+ naïve CD4 T cells, in this study). For this we utilized sorted immune cells from a cohort of donors that were previously genotyped and profiled for gene expression as part of the DICE study (n = 91; Database of Immune Cell Expression, Expression quantitative trait loci and Epigenomics)[8]. For 30 donors from DICE, we performed HiChIP experiments to profile interactions enriched for H3K27ac mark (Supplementary Data 1) to define contacts between active regulatory regions at high-resolution (5kb bin size) for naïve CD4 T cells (see Fig. 1A, B and Methods). Using our previously developed HiChIP loop caller FitHiChIP[64], we derive a total number of ~600k statistically significant HiChIP loops at 5kb resolution (see Methods; Supplementary Data 2, 3; Zenodo repository https://doi.org/10.5281/zenodo.13127086, file Complete_FitHiChIP_Loops_iQTL_Input.xlsx) across all 30 donors (union of loops). Among these 600k loops, 54% were identified in more than one donor with 40% shared among 2 to 10 donors (Supplementary Fig. 1A). Principal component analysis (PCA) (see Methods) with respect to the HiChIP contact counts did not reveal any outliers (Supplementary Fig. 1B), hence, we utilized data from all 30 donors for iQTL analysis.

Using ~600k FitHiChIP loops defined at 5kb resolution and considering bi-allelic SNPs within +/−1 bin of each loop anchor (15kb region per anchor; Fig. 1C), we then tested the association between SNP genotypes and loop strength as measured by HiChIP contact counts to map the interaction QTLs (or iQTLs) while accounting for covariates such as sequencing depth, sex, age group and race (see Methods). For this purpose, we employed RASQUAL, a Bayesian approach originally developed for mapping chromatin accessibility QTLs[65]. We started with the *default* model of RASQUAL that considers both genotype-dependent and allele-specific variation of contact counts. From the identified significant associations at 5% FDR, we applied a series of filters utilizing statistics from the *genotype-dependent* model, *allele-specific* model, paired *t*-test of allele-specific reads as well as concordance of trends for mean or median values (across donors) of HiChIP counts between genotype- and allele-specific estimates (see Methods, Fig. 1B, Supplementary Fig. 1C, D). This stringent filtering produced the final set of iQTL associations with 14,921 SNP-loop pairs, corresponding to 2292 loops and 9426 iQTL-associated SNPs (which we refer to as iQTLs when appropriate) across all the autosomal chromosomes (Supplementary Data 4). Among the 2292 HiChIP loops associated with the iQTL SNPs, the majority (1684 loops, ~73%) contain the iQTLs in their interacting anchors (5 kb) as opposed to the flanking bins that cover twice the size (+/− 5 kb) suggesting iQTL discovery is mainly driven by the assessment of exact loop anchors. To further characterize iQTL SNPs, we also assessed their overlap with enhancers predicted by activity-by-contact (ABC) model[66]. Around 40% (3765) of iQTLs were within 5 kb (14% or 1279 with exact overlap) of enhancer regions of the enhancer-to-gene (E2G) links predicted by the ABC model for the activated CD4 Naïve cells (see Methods).

We next assessed the utility of HiChIP loops associated with iQTLs, and found that ~41% of these loops are significant in more than 10 donors and only ~16% (375 out of 2292) are significant exclusively in one donor (Supplementary Fig. 2A). Further, 102 loops (~5%) were observed as significant in all the donors suggesting that there are iQTLs that strictly associate with the strength of the looping rather than its presence or absence. The 2292 iQTL-associated HiChIP loops also have higher median contact counts and are statistically significant (FDR < 0.01) in higher number of donors compared to the complete set of FitHiChIP significant loops, which are used as the input of iQTL analysis (Supplementary Fig. 2B). The iQTL associated loops, however, are of shorter distance range (mean distance: 140 kb, median distance: 50 kb) compared to the complete set of FitHiChIP loops (mean distance: 220 kb, median distance: 90 kb) suggesting increasing statistical power for detecting meaningful variation in contact counts for shorter distance ranges. Aggregate peak analysis[38,64] (APA) (see Methods) also confirmed higher enrichment of iQTL-associated loops compared to

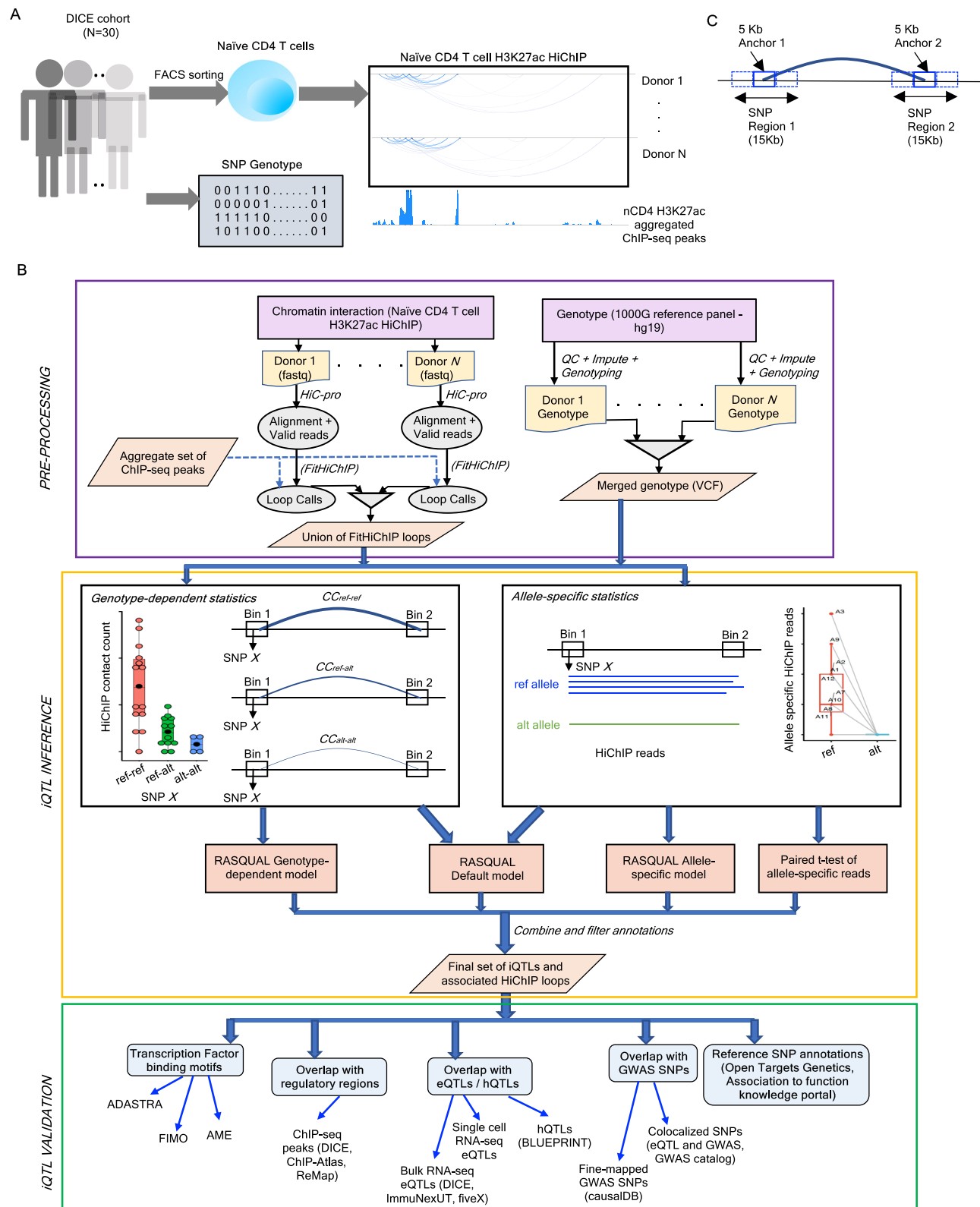

**Fig. 1 | Workflow of iQTL derivation. A** Schematic of the cohort, cell type, and data used in iQTL analysis. **B** Workflow of iQTL derivation from donor-wise HiChIP data together with genotype and ChIP-seq peak information. Union of HiChIP loops (derived from FitHiChIP) together with the donor-wise genotype information are used to run RASQUAL analysis. Final set of iQTLs (and associated HiChIP loops) are derived by combining outputs from three RASQUAL models and the paired *t*-test analysis of allele-specific reads. These iQTLs are further characterized and categorized with respect to their overlap with reference eQTLs, hQTLs, fine-mapped GWAS SNPs, colocalized SNPs (between eQTL and GWAS statistics), TF binding motif annotations, regulatory regions (e.g., ChIP-seq peaks) as well as reference SNP annotations. **C** Schematic of SNPs tested for association with one HiChIP loop. For each loop between two 5kb anchors, SNPs within the 15 kb region surrounding each anchor (+/−1 bin from the anchor) are tested for association with the HiChIP loop. QC quality control, CC HiChIP contact counts, VCF variant calling format, eQTL expression quantitative trait loci, hQTL histone quantitative trait loci.

the complete set of FitHiChIP loops with respect to either the background HiChIP contacts (Supplementary Fig. 2C) or the CD4 T cell type merged donor Hi-C contacts provided in Shi et al.[59] (Supplementary Fig. 2D). Pathway analysis (see Methods) reveals that the genes overlapping the anchors (interacting bins) of these iQTL-associated loops are highly relevant to the immune system processes and cell signaling among other biological processes (Supplementary Fig. 2E).

To assess the overlap between iQTLs and more established QTL measurements such as eQTLs and histone modification QTLs (hQTLs), we utilized two studies profiling eQTLs in nCD4 T cells (DICE[8] and ImmuNexUT[7]) and H3K27ac hQTLs from Blueprint WP10 Phase II data[63]. We observed that ~64% (5933 out of 9426) iQTL SNPs overlap with eQTLs or hQTLs (Fisher's exact test $p$-value < 0.00001, OR = 1.43; see Methods; Supplementary Data 11A) while the remaining ~36% iQTL SNPs (3433 out of 9426) do not (Supplementary Fig. 3A, B). iQTLs common with hQTLs or eQTLs, however, showed high concordance among their effect sizes (iQTLs - H3K27ac hQTLs: Pearson correlation = 0.78, $p$-value < $2.2e^{-16}$; iQTLs - DICE nCD4 eQTLs: Pearson correlation = 0.65, $p$-value < $2.2e^{-16}$) (Supplementary Fig. 3C, D), suggesting that the changes in corresponding HiChIP contact counts are related with the H3K27ac levels and gene expression. To further validate whether iQTL effects are explained by the reference eQTLs or hQTLs, we accounted for LD ($R^2$ > 0.8) between iQTL SNPs and these QTLs and observed that ~28% (2603 out of 9426) iQTLs are not in LD with these reference QTLs and only ~38% of iQTLs are in LD with the BLUEPRINT hQTLs (Supplementary Fig. 3E). Further, only for ~31% iQTL-associated HiChIP loops, corresponding iQTL SNPs overlap or exhibit strong LD ($R^2$ > 0.8) with hQTLs where the hQTL-associated peaks overlap with one of the iQTL loop anchors (Supplementary Fig. 3F). Overall, iQTL-associated HiChIP loop anchors showed higher overlap with reference CD4 naïve DICE eQTLs or hQTLs compared to either FitHiChIP loop anchors not associated with iQTLs (OR: 1.39, Fisher's exact test $p$-value < 0.00001) or random anchors (OR: 5.14, Fisher's exact test $p$-value < 0.00001) (see Methods, Supplementary Data 11C). The same was true when we compared between the ChIP-seq peaks in the iQTL associated HiChIP loop anchors versus the rest, in terms of their overlap with reference eQTLs or hQTLs (OR: 2.46, Fisher's exact test $p$-value < 0.00001) (see Methods, Supplementary Data 11D). These results indicate that a significant fraction of iQTLs is driven by the variation of 3D chromatin conformation without underlying changes in histone modification profiles and some without corresponding changes in gene expression in the profiled cell type.

## iQTLs overlapping with eQTLs in naïve CD4 T cells are enriched for fine-mapped GWAS SNPs and TF binding motifs

To understand the properties of iQTL-associated SNPs, we first tested their overlap with the reference nCD4 T cell eQTLs provided in two different eQTL databases for immune cell types, namely DICE[8] and ImmuNexUT[7]. We observed that 5381 iQTLs (~57%) are also eQTLs of nCD4 (Fisher's exact test $p$-value < 0.00001, OR: 1.88, see Methods; Supplementary Data 11B) with respect to either of these two databases (Supplementary Fig. 3G and Supplementary Data 4). In total, 1276 out of 2292 iQTL loops (~56%) are associated with these 5381 iQTLs which are also nCD4 eQTLs (Supplementary Figs. 2F, 3G, and Supplementary Data 4). As the HiChIP protocol returns high-resolution regulatory contact maps[47], we reasoned that iQTLs should also be enriched for regulatory SNPs. Indeed, we observed that 1725 iQTLs (~32%) of this category fall within the regulatory regions defined by aggregate nCD4 H3K27ac ChIP-seq peaks, and ChIP-seq peaks for various CD4 T cell subsets and regulatory TFs provided in the ChIP-Atlas[67] database (see Methods and Supplementary Data 4), and such a fraction of regulatory SNPs is substantially higher (OR: 3.16; Fisher's exact test $p$-value < 0.00001; Supplementary Data 11E) than that of the reference DICE nCD4 eQTLs (~13% regulatory SNPs) (Supplementary Data 5). Out of the 5381 iQTLs in this category, 2163 (~40%) were within 5kb of ABC

enhancers (725 or ~13.5% with exact overlap)[66]. We next checked whether iQTLs are enriched for transcription factor (TF) binding motifs (see Methods) supported by various techniques such as motif scanning by FIMO[68,69], allele-specific motif enrichment by AME[70] or evidence of allele-specific binding from ChIP-seq data by ADASTRA[71,72] (https://adastra.autosome.org/). Considering iQTLs which are also nCD4 eQTLs, ~62% SNPs overlapped with TF binding sites derived by at least one of these approaches (compared to ~51% DICE Naïve CD4 eQTLs; OR: 1.58; Fisher's exact test $p$-value < 0.00001; Supplementary Data 11F), out of which ~19% SNPs showed TF binding motif evidence from at least two approaches (compared to 1.3% DICE Naïve CD4 eQTLs; OR: 17.3; Fisher's exact test $p$-value < 0.00001; Supplementary Data 11F; Supplementary Fig. 4A). Allele-specific motif enrichment analysis by AME on these iQTLs (see Methods) revealed the enrichment of regulatory TFs such as TCF3 (an E protein that plays essential roles in B and T cell development) and FOS (a subunit of the AP-1 TF that plays role in cell proliferation, differentiation and in response to stimuli such as cytokines) for all motif databases (JASPAR, cisbp and HOCOMOCO) and the TFs STAT3 (a lineage defining TF for Th17 and regulates B and T cell responses to stimuli), and REL (a member of NFKB family of TFs associated with ulcerative colitis and rheumatoid arthritis) for two motif databases (Supplementary Fig. 4B, Supplementary Data 4). De novo motif analysis using FIMO showed significant enrichment of regulatory TFs YY1, CTCF, IRF1 (interferon regulatory factor 1), SP1 (zinc finger TF, binds many promoters) and SP2 in all motif databases, and TFs STAT1 (response to pathogens, interferon signaling, Th1 differentiation), MYC (oncogene), ETS1 (a member of ETS family of transcriptional activators) in two motif databases (Supplementary Fig. 4C, Supplementary Data 4). These results suggest that iQTLs overlap binding sites of TFs that are important for T cell differentiation and function as well as TFs implicated in chromatin loop formation such as CTCF and YY1.

To show that iQTLs prioritize putative causal or disease-specific eQTLs, we next assessed iQTLs and DICE nCD4 eQTLs by their overlap with fine-mapped GWAS SNPs from various immune diseases (Supplementary Data 6) provided in the CausalDB[73] database (http://www.mulinlab.org/causaldb/browse.html). We observed that the iQTLs shared with eQTLs show significantly higher overlap (assessed by Fisher's exact test) with the GWAS SNPs compared to the full set of eQTLs, particularly for the diseases Asthma, Crohn's disease, Diabetes Mellitus, Allergic Rhinitis ($p$-value < 0.00001), psoriasis ($p$-value < 0.001), Ulcerative colitis, T1D, IBD, and multiple sclerosis ($p$-value < 0.05) (Supplementary Fig. 4D). In fact, the full set of iQTLs (eQTLs or not) also exhibit higher overlap with the fine-mapped GWAS SNPs for the diseases Asthma, Crohn's disease ($p$ < 0.00001), psoriasis ($p$ < 0.001), Diabetes Mellitus, multiple sclerosis ($p$ < 0.05) compared to the reference DICE nCD4 eQTLs (Supplementary Fig. 4D). In addition, we performed heritability analysis, which gives insight of the genetic contribution to the complex traits or diseases[74]. We compared iQTLs and DICE nCD4 eQTLs by their heritability enrichment for various immune diseases (Supplementary Data 7), by adapting the stratified LD score regression[75] (S-LDSC) method to compute the heritability enrichment for a given set of SNPs (see Methods and https://github.com/ay-lab/S_LDSC_SNP). We observed that the complete set of iQTLs as well as iQTLs overlapping nCD4 eQTLs provided higher heritability enrichment and statistical significance compared to the complete set of eQTLs for the IBD, IBS, Crohn's disease, and T2D. For asthma and SLE, however, the overall set of DICE eQTLs produced higher heritability enrichment (Supplementary Fig. 4E). Overall, these results suggest that iQTLs prioritize eQTLs associated with disease risk and higher enrichment of heritability for immune-related GWAS traits.

Among the 1276 iQTL loops associated with iQTLs that are also nCD4 eQTLs, a high fraction (884 loops or ~69%) are only associated with such types of iQTLs (Supplementary Fig. 2F and Supplementary Data 4). An example of such an iQTL loop is a 40 kb loop between the

genes *IKZF3* and *GSDMB* within the 17q21.1 locus, a locus implicated in many immune-associated diseases (Schmiedel et al. 2016). We obtained 24 iQTLs (Supplementary Data 4A) associated with this loop (Fig. 2A), all of them being significant nCD4 eQTLs of *GSDMB* and 21 of them (like the SNPs rs2305479, rs7216389) are nCD4 eQTLs for the asthma risk associated gene *ORMDL3* (Supplementary Fig. 5A). Genotype-dependent variation of HiChIP contact counts for these iQTLs show similar trends with gene expression, but do not exhibit significant genotype-dependent or allele-specific variation of 1D ChIP or HiChIP coverages (Fig. 2B, Supplementary Fig. 5A). These iQTLs are also relevant to various immune diseases - all 24 iQTLs belong to the fine-mapped (95% credible sets) GWAS SNPs (CausalDB[73] database) for the trait Asthma; a few belong to the fine-mapped GWAS SNPs for the traits Type 1 Diabetes (T1D), Systemic Lupus Erythematosus (SLE), and Ulcerative Colitis (Supplementary Data 6). Further, 16 out of 24 iQTLs belong to the list of PICS[76] fine-mapped SNPs for five different immune diseases (Supplementary Data 4A) as provided in ref. [10]. The SNP rs56380902 (chr17: 38066372) is also shown to be associated with time-dependent activation of *ORMDL3* in the single-cell eQTL study of memory T cells[13]. Another SNP rs56750287 (chr17:38062944) belongs to the list of expression-modulating variants (emVars), a set of regulatory GWAS SNPs validated by MPRA having potential regulatory effects in Jurkat T cells[10] (Supplementary Data 4A). Although all 24 iQTLs are in a tight haploblock ($R^2 > 0.8$) they are not linked to the lead eQTL rs17608925 (chr17:38082831, not an iQTL) of the genes *GSDMB* and *ORMDL3* (Supplementary Fig. 5B) suggesting that gene expression variation may not have a 1-to-1 relationship with changes in chromatin looping. In support of this, we observed that the two SNPs we previously identified to overlap with CTCF binding sites in naïve CD4 T cells[77], namely rs4065275 (chr17:38080865) and rs12936231 (chr17:38029120) were also in tight LD with iQTLs but not the lead eQTL (Supplementary Fig. 5B). These two SNPs were shown to switch the CTCF binding from the gene *ZPBP2* to *ORMDL3* intronic region in the risk haplotype (G for rs4065275 and C for rs12936231) compared to non-risk alleles (A for rs4065275 and G for rs12936231).

Overall, iQTLs common to DICE nCD4 eQTLs (2471 iQTLs) are associated with 275 eGenes. Fine mapping of these eQTLs using the tool FINEMAP[20] (see Methods) produced 111 fine-mapped eQTLs associated with 84 eGenes (~31% of 275 eGenes), suggesting that a high fraction of eGenes show correlated changes in gene expression and HiChIP contacts. An example of such an iQTL and a fine-mapped eQTL is rs3761847 (chr9:123690239), associated with a 50kb HiChIP loop between the genes *TRAF1* and *PHF19* (Supplementary Fig. 5C and Supplementary Data 4B), showing significant variation of allele-specific HiChIP contacts and 1D HiChIP reads but not for 1D ChIP-seq coverage (Supplementary Fig. 5D), and is also a lead eQTL of *TRAF1*. The *TRAF1* gene is a member of the TNF receptor (TNFR) associated factor (TRAF) 1, which is an intracellular signaling protein involved in the regulation of immune responses, T cell activation, differentiation, and survival. Specifically, *TRAF1* is highly expressed for the RA patients, while the gene *PHF19* (PHD finger protein 19) is downregulated for the RA patients[78]. The iQTL rs3761847 overlaps with the reference CD4 Naïve H3K27ac and H3K4me3 ChIP-seq peaks, and also belongs to the list of PICS fine-mapped SNPs (Supplementary Data 4B) for the trait Rheumatoid arthritis (RA)[10]. In fact, its association with RA is also confirmed by the GWAS catalog[1,79], Phenome-wide association study (PheWAS) (https://a2f.hugeamp.org/), and Open Targets Genetics framework[80,81]. A previous study[82] highlighted that the allele G of the intronic SNP rs3761847 (chr9:123690239:G:A) is the risk allele and this allele is also associated with higher expression of *TRAF1* (eQTL in nCD4 in both DICE and ImmuNexUT; Supplementary Fig. 5E). Interestingly, the G allele was associated with a weaker loop from the *TRAF1* promoter region (Supplementary Fig. 5D) and 7 other iQTL SNPs that are within 15 kb distance of rs3761847 and are in near perfect LD were all also associated with weaker loops in the region (Supplementary Data 4B). Upon examination

none of these seven iQTLs overlapped CTCF binding motifs but rs3761847 overlapped a motif for ZNF263, another zinc finger protein (ZNF) the binding sites of which are recently shown to be enriched at relocated cohesin upon CTCF and MAZ depletion in mouse embryonic stem cells[83]. Our motif enrichment analysis using FIMO, also highlighted ZNF263 binding sites as one of the motifs enriched for iQTLs that are also nCD4 eQTLs (Supplementary Fig. 4C, Supplementary Data 4), suggesting a potential insulator role for this ZNF protein in T cells.

Similar to the iQTL rs3761847 which is also a lead DICE nCD4 eQTL of *TRAF1*, overall, we obtained 384 iQTLs which are also the lead eQTLs of one or more eGenes in either DICE or ImmuNexUT (Supplementary Data 8). As an example, all 10 iQTLs associated with the 40kb HiChIP loop in the *CD247* locus (Supplementary Data 4C and Fig. 2C) are also the lead eQTLs ($p$-value = $6.8 * 10^{-7}$) of the *CD247* gene (ImmuNexUT), and they are mutually in strong to moderate LD ($R^2 > 0.7$) (Supplementary Fig. 5F). The *CD247* encodes for the CD3 zeta chain and is part of the T-cell-receptor (TCR)-CD3 complex and hence critical in transmission of TCR-mediated signals in T cells. Genotype dependent trends of HiChIP contact counts for these 10 QTLs (e.g., rs2949661) are concordant with their eQTL trends of *CD247* expression (Fig. 2D, Supplementary Fig. 5G) for different CD4 T cell subsets. Similar trends are observed for allele-specific ChIP-seq (statistically significant; $p$-value = 0.001, paired $t$-test) and 1D HiChIP coverage (Fig. 2D). All 10 iQTLs are also fine-mapped GWAS SNPs (CausalDB[73] database) for the traits asthma and allergy (Supplementary Data 6). The SNP rs2949661 (chr1:167424924) in particular, is a regulatory SNP overlapping with reference nCD4 H3K27ac ChIP-seq peaks, ChIP-seq peaks for the TFs ETS1, BCL6, RUNX1, BRD4 and RNA-pol-II (ChIP-Atlas[67]) (Supplementary Data 4C). This SNP rs2949661 also overlaps with activated CD4 Naïve cell ABC enhancers[66] associated with the gene *CD247* via significant E2G links (Fig. 2C). Further, rs2949661 exhibits allele-specific binding of *ETS1* (FDR = $7.2 * 10^{-4}$ as per the ADASTRA[71,72] database) in Th1 cells. Lastly, rs2949661 is also shown to be associated with Sjogren's syndrome by being an eQTL to both genes *CD247* and cellular repressor *CREG1* and by altering TAD interactions involving *CD247*[84]. Another iQTL rs2995091 (chr1:167434277) in near perfect LD with rs2949661 overlaps binding motif of IRF1 (activator of interferon alpha and beta transcription) (Fig. 2E), which is another TF with motifs enriched in this category of iQTLs (Supplementary Fig. 4C, Supplementary Data 4). We also obtained rs2995091 as one of the nCD4 eQTLs from a recent single-cell eQTL study[15]. Open Targets Genetics[80,81] confirms promoter capture Hi-C loops[45] between these iQTLs and *CD247*. Overall, our results suggest that, for some loci, iQTLs may find a refined subset of eQTLs where the effect on gene expression can be attributed to underlying changes in chromatin contacts. In many cases, this is likely mediated by changes in TF binding to the iQTL SNP or to other nearby SNPs that are in LD. The concordance between iQTL and eQTL effects may depend on which TFs are binding weaker/stronger to these SNPs. This iQTL subset is also enriched in fine-mapped GWAS SNPs for various immune diseases as supported by genome-wide analysis and the example loci highlighted above supporting their significance.

## Naïve CD4 T cell iQTLs pinpoint cell-specific eQTLs in other CD4 T cell subsets, revealing potential regulation of gene expression by chromatin interactions

Understanding the variation of eQTL effects according to different cell states or differentiation stages such as different CD4 T cell subsets[11] or immune cell types[15] or inferring activation or differentiation-dependent eQTLs[12,77,85] help reveal the impact of genetic effects on complex traits. In view of this, we next cataloged 1143 (~12%) iQTLs that are not eQTLs in nCD4 but are eQTLs for other CD4 T cell subsets (8 cell types including stimulated CD4 T cells, Tfh, Th1, Th2, Th17, Th1/17, naïve and memory Tregs compiled from DICE and ImmuNexUT; see Methods, Supplementary Fig. 3G, Supplementary Data 4). In total, these 1143 iQTLs were associated with 482 distinct HiChIP loops.

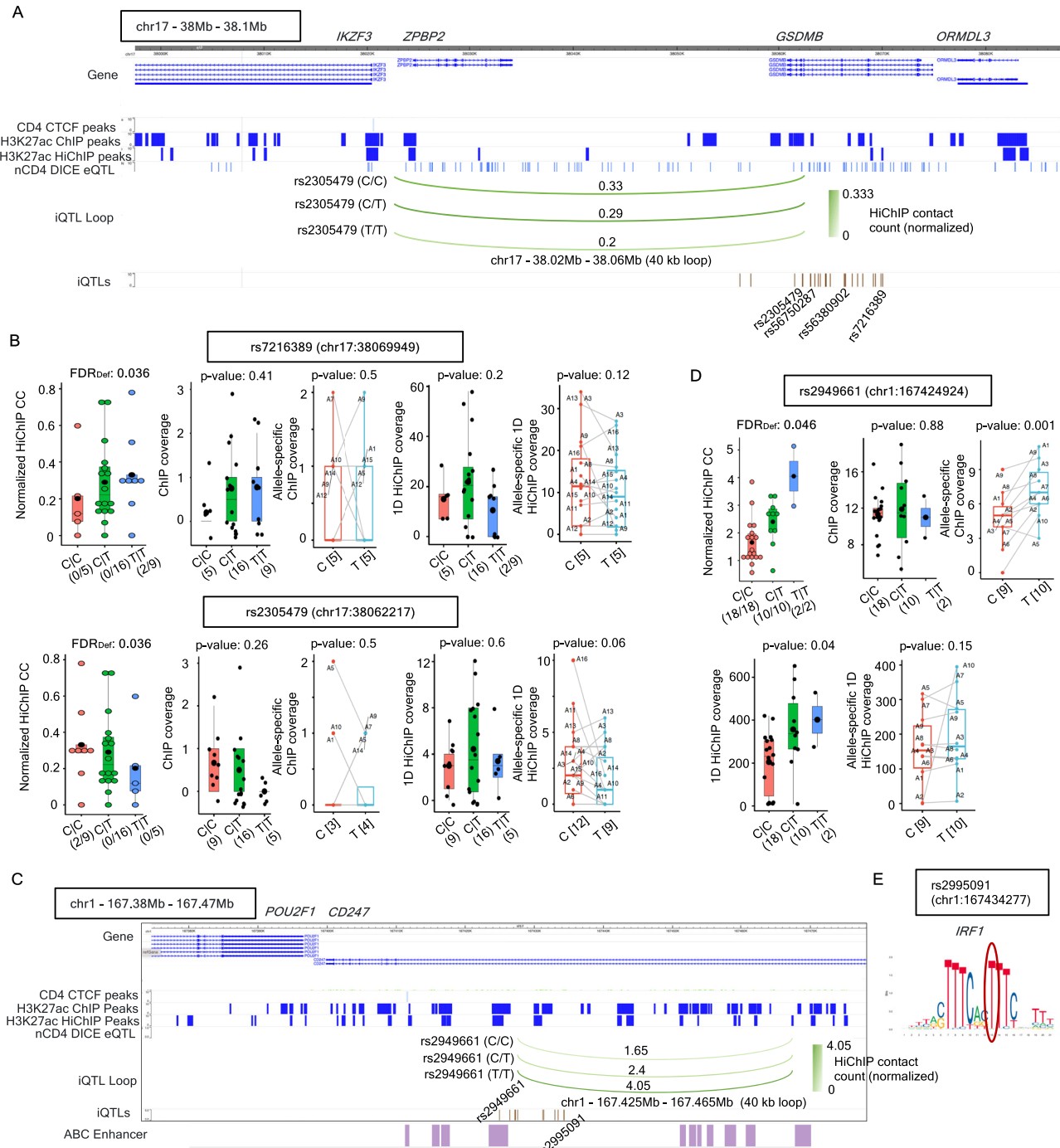

**Fig. 2 | Examples and properties of iQTLs which are also eQTLs in Naïve CD4 T cells. A** Example of iQTL SNPs (which are also nCD4 eQTLs) associated with a 40kb HiChIP loop in chr17 *ORMDL3* locus. The color scale of green arcs and the values on top of them indicate mean normalized HiChIP contact counts (indicated by numbers) by genotype for one of these SNPs rs2305479. **B** Trends of genotype-dependent sequencing depth normalized HiChIP contact counts, ChIP-seq coverage, 1D HiChIP coverage, and allele-specific variation of ChIP and 1D HiChIP reads, for two iQTLs rs7216389 (top) and rs2305479 (bottom) associated with the 40kb HiChIP loop in **A**. For genotype dependent trends, X axis indicates different SNP genotypes, numbers in the formats (*a* / *b*) or (*b*) denote that the corresponding genotype is present in *b* donors, out of which *a* donors have this HiChIP loop as significant (by FitHiChIP). For allele-specific trends, X axis denotes different alleles, and numbers in the format [*c*] indicate that *c* heterozygous donors have nonzero reads for this allele. FDR_Def denotes the statistical significance (FDR) of this iQTL SNP-loop pair using the default RASQUAL model. For

genotype-specific plots, *p*-values are obtained from linear regression (ANOVA) while for allele-specific plots, *p*-values are computed by one-sided paired *t*-test. Boxplots contain mean (bigger black dots), median (middle lines), 25th, 75th percentiles (box) and individual samples (small circles or dots or symbols A* where * indicates numbers). **C** Example of iQTLs associated with a 40kb HiChIP loop in the *CD247* locus, such that the iQTLs are also the lead eQTLs of *CD247* in nCD4 (ImmuNexUT). Green arcs indicate mean normalized HiChIP contact counts by genotype for one of these SNPs rs2949661. Enhancers reported by the E2G links of the activity-by-contact (ABC) model for the activated naïve CD4 T cell type are also shown. **D** Similar to **B**, for the iQTL rs2949661 associated with the loop in **C**. **E** TF motif for *IRF1* overlapping the SNP rs2995091 (according to FIMO). The circle denotes the position of the SNP/iQTL within the TF binding motif. nCD4: Naïve CD4, CC: contact count. Source data are provided as a Source Data file.

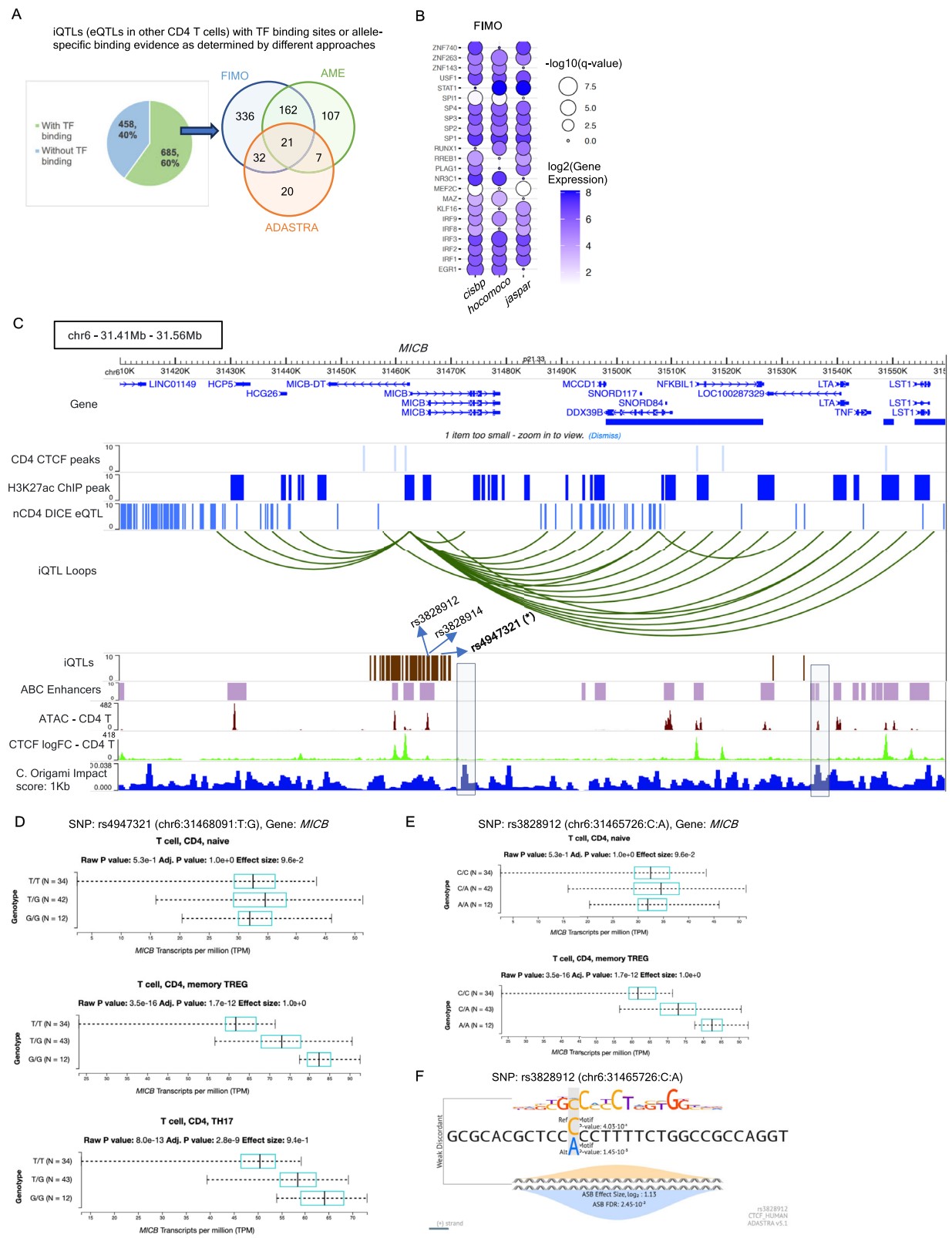

Around 34% (386) of these iQTLs overlapped with regulatory peaks (Supplementary Data 4), a percentage comparable to iQTLs which are nCD4 eQTLs, and much higher (OR: 3.41; Fisher's exact test p-value < 0.00001; Supplementary Data 11E) compared to the complete set of DICE nCD4 eQTLs. Overall, 474 iQTLs (~41%) of this category overlap are within 5kb of ABC enhancers (198 or ~17% with exact overlap) of activated CD4 Naïve T cells. Around 60% of iQTLs in this category

overlapped TF binding motifs supported by at least one of the techniques FIMO, AME or ADASTRA TF (OR 1.45, Fisher's exact test p-value < 0.00001, compared to the complete set of DICE nCD4 eQTLs, Supplementary Data 11F) and ~20% had motif overlap supported by at least two methods (OR 18.5; Fisher's exact test p-value < 0.00001, compared to the complete set of DICE nCD4 eQTLs, Fig. 3A, Supplementary Data 4, Supplementary Data 11F), both percentages similar to iQTLs

**Fig. 3 | Properties of iQTLs which are not eQTLs in nCD4 T cell but are eQTLs in other CD4 T cell subsets. A** iQTLs in this category with or without TF binding motifs (left); overlap of iQTLs with TF binding motifs from different approaches – de novo motif analysis using FIMO, allele-specific motif enrichment by AME, and ADASTRA TF database. For AME and FIMO, iQTLs overlapping TF binding motifs with respect to one or more of the three databases (cisbp, hocomoco or JASPAR) are considered. **B** TF motif enrichment by FIMO for these iQTLs, such that the motif is supported by at least two of the three motif databases cisbp, hocomoco or JASPAR. TFs with gene expression > 1 TPM having motifs with *p*-value < 1e-6 are plotted. **C** *MICB* locus having multiple iQTLs associated with multiple HiChIP loops, where the iQTLs are eQTLs of *MICB* in T helper subsets including Th1, Th2, Th17, Tfh and Tregs. The symbol * indicates that the corresponding SNP (rs4947321) is a fine-mapped eQTL of *MICB* in Treg memory (DICE database). ABC model enhancers,

corresponding to the E2G links of the activated naïve CD4 T cell type, are also shown. Bottom three tracks show CD4 T cell ATAC-seq, CTCF log fold change over control ChIP-seq track (used as an input to C. Origami) and the impact score at 1 kb resolution obtained by C. Origami, where higher impact score indicates higher impact on 3D chromatin organization by perturbation of the corresponding loci. **D** Genotype-dependent variation of *MICB* expression for the SNP rs4947321 in the cell types CD4 Naïve, Treg Memory and Th17 (according to the DICE database). Boxplots indicate median (middle lines), 25th, 75th percentiles (box) and minimum and maximum values (whiskers). **E** Similar to **D**, for the SNP rs3828912 in the cell types Naïve CD4 and Treg Memory. **F** CTCF binding motif overlapping with the SNP rs3828912 in Spleen, according to the ADASTRA database. TF Transcription factor, nCD4 Naive CD4. Source data are provided as a Source Data file.

that are nCD4 eQTLs (Supplementary Fig. 4A). These iQTLs, similar to iQTLs that are nCD4 eQTLs, showed enrichment of TF motifs (FIMO analysis) for zinc finger proteins such as ZNF263, SP1/2/3/4 and MAZ, transcriptional activators STAT1 and RUNX1, as well as interferon response factors IRF1/2/3 but not for CTCF or YY1 (Fig. 3B, Supplementary Data 4). Allele-specific enrichment analysis using AME also identified additional TF motifs such as ZEB1 (regulates T cell development differentiation) and REL (Supplementary Fig. 6A, Supplementary Data 4).

To understand whether these iQTLs and the genotype-dependent variation of the associated loops in nCD4 are mechanistically linked to eQTL effects in other CD4 T cell subsets, we focused on the differentially expressed genes (DEGs; FDR < 0.05) upregulated (log2 fold change > 1) in various CD4 T cell subsets compared to nCD4 cells. Overlapping the considered set of iQTLs with eQTLs of these DEGs, we obtained 268 such iQTLs and 71 DEGs (Supplementary Data 9). An example DEG upregulated in Treg memory cell type is the *MICB* (MHC class I polypeptide-related sequence B) gene, which is a ligand for the *NKG2D* receptor. This *MICB* locus contains multiple HiChIP loops associated with a total of 90 iQTLs where none of them are eQTLs of nCD4 in either DICE or ImmuNexUT (Fig. 3C). However, 43 of these 90 iQTLs are eQTLs of *MICB* for Treg memory cells (Supplementary Data 4D). To showcase the importance of this *MICB* locus in regulating 3D genome organization for CD4 T cells, we applied the deep learning technique C. Origami[86] which models 3D contact map from DNA sequence and cell-type-specific epigenomic tracks such as ATAC-seq and CTCF fold changes (see Methods). Using the pre-trained model, C. Origami also supports screening a specific region or locus by a fixed resolution (like 1Kb) to assess the potential perturbation impact of regulatory elements on 3D genome. Using a pre-trained C. Origami model on CD4 T cell Hi-C dataset provided in ref. 59 (see Methods), we applied perturbation screening of 1 kb resolution and observed that the locus containing iQTLs also exhibit high impact scores (indicating higher regulatory impact), thereby potentially impacts 3D genomic changes (Fig. 3C). The iQTL rs4947321 (chr6:31468091), in particular, is a fine-mapped eQTL of *MICB* (derived by FINEMAP[20]) in Treg Memory as well as an eQTL for various other CD4 T cell subsets including Tfh, Th1, Th2 and Th17, all with higher *MICB* expression for the non-reference allele G (Fig. 3D–E). This iQTL is in strong LD (R² > 0.8) with 38 out of the remaining 42 iQTLs, which are also eQTLs of *MICB* in Treg Memory (Supplementary Fig. 6B). Two of such linked iQTLs rs3828912 and rs3828914, showing similar trends of *MICB* expression and HiChIP contacts (higher values for non-reference alleles) (Fig. 3E), overlap with T-cell-specific CTCF ChIP-seq binding sites[87], and CTCF ChIP-seq peaks from spleen (ReMap DNA binding database[88]). These SNPs also overlap with the enhancers involved in the E2G links to the gene *MICB*, according to the ABC model[66] (Fig. 3C). The SNP rs3828912 also shows allele-specific binding preference for CTCF for spleen and for pancreas tissue in the ADASTRA database (Fig. 3F). However, these iQTLs do not show significant genotype-dependent or allele-specific variation of 1D ChIP or HiChIP coverages (Supplementary Fig. 6C) suggesting that the

effects of these SNPs are solely at 3D level. Multiple other iQTLs in this locus are GWAS SNPs for various immune diseases, such as Systemic lupus erythematosus or SLE (rs4947321, rs6930510, rs9267366, rs2534671, rs2534681), Multiple Sclerosis (rs3828912), Type 1 diabetes or T1D (rs4947321, rs2534671, rs2855807, rs2534681), and Rheumatoid Arthritis or RA (rs5027463) according to PheWAS (https://a2f.hugeamp.org/) and causalDB databases (Supplementary Data 6). These results suggest that iQTLs that are not eQTLs in the studied cell type may likely have underlying eQTL effects that will surface when the specific cell type is perturbed, activated or differentiated into different fates.

## iQTLs not eQTLs in any CD4 T cell subset are enriched for fine-mapped GWAS SNPs

The third category of iQTLs constitute 2902 SNPs (~31%) which are not eQTLs (either DICE or ImmuNexUT) in any CD4 T cell subset that is profiled (Supplementary Data 4). In terms of overlap with regulatory regions (~30%) (OR: 2.87, Fisher's exact test *p*-value < 0.00001 with respect to the complete DICE nCD4 eQTLs, Supplementary Data 11E), and TF binding motifs (FIMO analysis: 61% with 20% supported by two databases, odd ratios of 1.51 and 18.45, respectively, with respect to the complete DICE nCD4 eQTLs; Supplementary Data 11F), this set of iQTLs were very similar to the two previously discussed categories above that are eQTLs in either nCD4 T cells or other CD4 subsets (Fig. 4A, Supplementary Data 4). In terms of enriched motifs from FIMO and allele-specific analysis using AME, the identified TFs were also very similar (e.g., ZNFs, IRFs, SP1/2/3/4) with YY1 coming up as significant from both approaches but without an enrichment for CTCF (Supplementary Fig. 7A, B). Out of 696 HiChIP loops solely associated with iQTLs which are not eQTLs in any CD4 T cells (Supplementary Fig. 2F), an example is a 60kb loop associated with 33 iQTLs within the *RUNX3* locus (Fig. 4B and Supplementary Data 4E). These iQTLs are mutually in tight (R² > 0.8) LD (Supplementary Fig. 7C), and they do not exhibit any genotype or allele-specific changes with respect to 1D ChIP-seq or HiChIP coverage (Fig. 4C). The *RUNX3* gene is a transcriptional activator and regulates the differentiation of CD4+ T cells into various subsets (Th1, Treg) and is expressed across naïve cells and different T helper populations in human. Although not eQTLs in CD4 T cell subsets, majority of these iQTLs are associated with T1D and Atopic Dermatitis, as confirmed by CausalDB fine-mapped GWAS[73], PICS[76] fine-mapped SNPs provided in ref. 10 (Supplementary Data 4E, 6), and PheWAS studies (https://a2f.hugeamp.org/). These iQTLs, being proximal to the gene *RUNX3*, also exhibit high variant-to-gene (V2G) association with the gene, as reported by Open Targets Genetics platform[80,81]. The iQTLs rs10751776 and rs11249221, in particular, are eQTLs in myeloid dendritic cells (mDC) (Supplementary Fig. 7D, E). The rs10751776 SNP shows CTCF binding motifs according to both ADASTRA database and FIMO (Fig. 4D), while rs11249221 overlaps a binding motif for ZEB1 (Fig. 4E). These results suggest that iQTLs with no associated eQTLs in any CD4 cell subset may also be of interest in annotating GWAS variants that cannot be explained by other QTL modalities. The limited

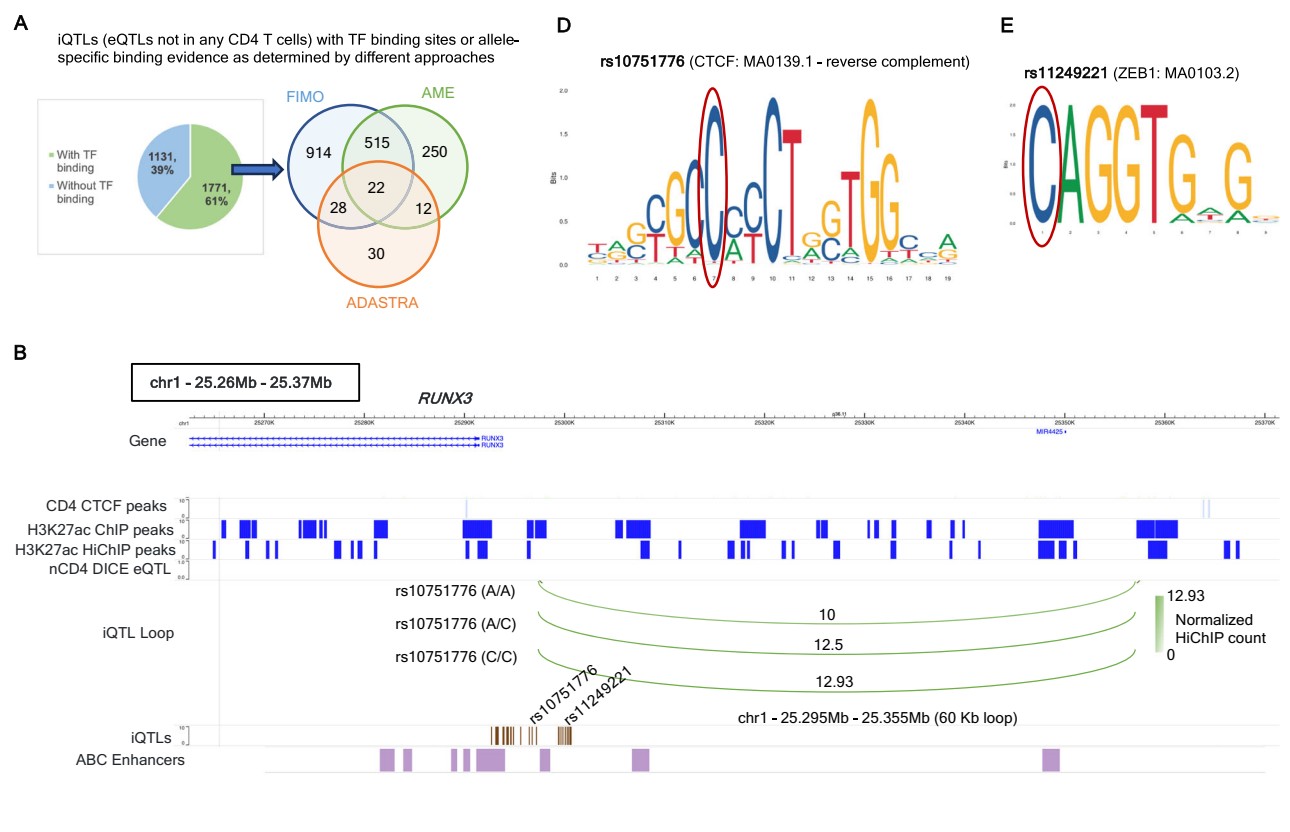

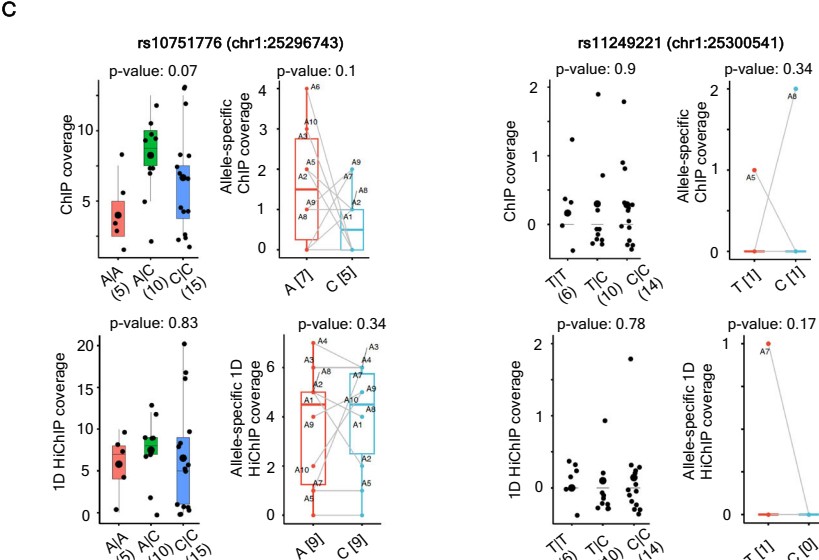

**Fig. 4 | Example of iQTLs which are not eQTLs in any CD4 T cell subsets. A** iQTLs that are not eQTLs in any CD4 subset with or without TF binding motifs (left); overlap of iQTLs with TF binding motifs from different approaches – de novo motif analysis using FIMO, allele-specific motif enrichment using AME, and ADASTRA TF database. For AME and FIMO, iQTLs showing TF binding motifs with respect to any of the three databases cisbp, hocomoco or JASPAR are considered. **B** Example iQTLs rs10751776 and rs11249221 associated with a 60kb HiChIP loop involving the gene *RUNX3*. Green arcs indicate mean normalized HiChIP contact counts by genotype for rs10751776. ABC model enhancers, corresponding to the E2G links of the activated naïve CD4 T cell type, are also shown. **C** Trends of genotype-dependent ChIP-seq coverage, 1D HiChIP coverage, and allele-specific variation of ChIP and 1D HiChIP reads, for two iQTLs rs10751776 (left) and rs11249221 (right) associated with the 60kb HiChIP loop in **B**. For genotype dependent trends, X axis

indicates different SNP genotypes, numbers in the format (*b*) denote that the corresponding genotype is present in *b* donors. For allele-specific trends, X axis denotes different alleles, and numbers in the format [*c*] indicate that *c* hetero-zygous donors have nonzero reads for this allele. For genotype-specific plots, *p*-values are obtained from linear regression (ANOVA) while for allele-specific plots, *p*-values are computed by one-sided paired *t*-test. Boxplots show median (middle lines), 25th, 75th percentiles (box) and individual samples (black dots or dots with symbols A* where * indicates numbers). **D** CTCF binding motif overlapping the SNP rs10751776 according to the JASPAR database (reverse complement). Note that this region does not have any CTCF binding (ChIP-seq signal) in naïve CD4 T cells. **E** The SNP rs11249221 overlaps ZEB1 binding motif, according to the JASPAR database. CC: contact count. Source data are provided as a Source Data file.

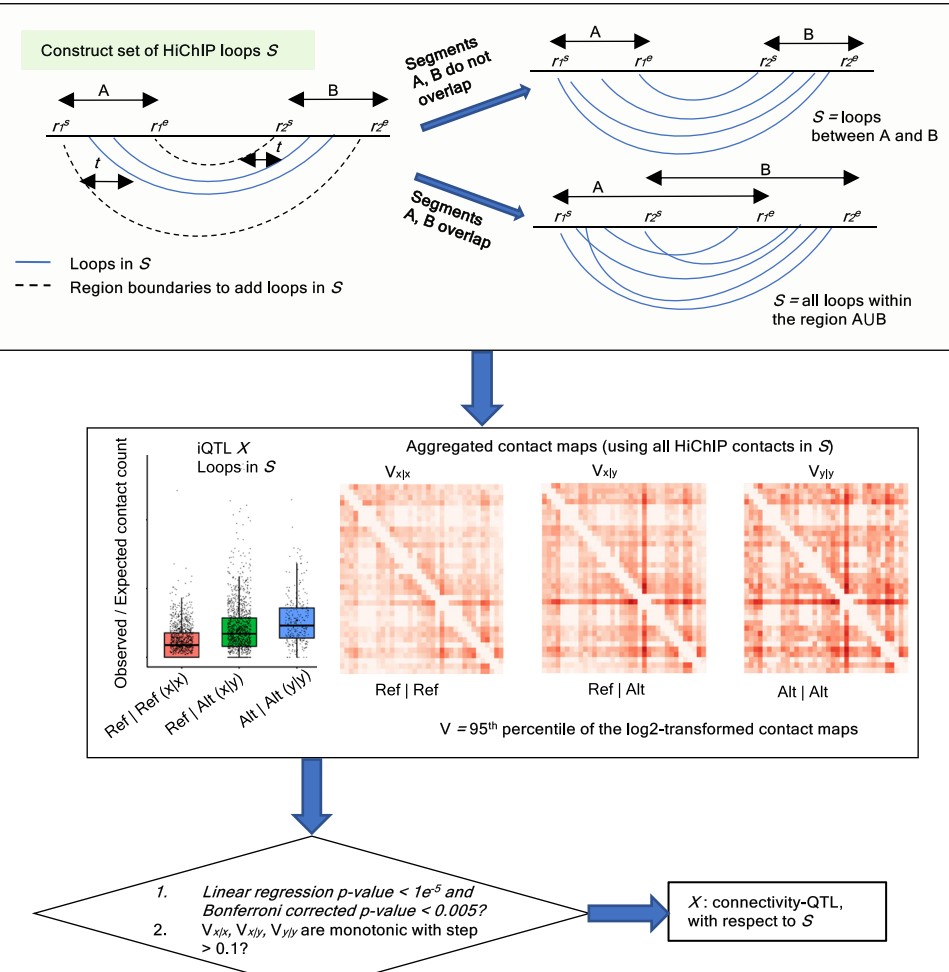

**Fig. 5 | Workflow to derive connectivity-QTLs.** For a given iQTL $X$ and the associated HiChIP loop $L$, connected component labeling algorithm is adapted to derive the set of HiChIP loops $S$ spatially proximal to the given loop $L$ (inclusive). The union of interacting bins A and B for the set of loops $S$ can either overlap or not. To decide whether $X$ is a connectivity-QTL associated with $S$, observed / expected contact counts for all loops in $S$ are combined per genotype and the trend is tested for statistical significance with additional filtering criteria on the differences between contact maps aggregated per genotype. Here, $r_1^s$ and $r_1^e$ denote the start and endpoints of the region A, while $r_2^s$ and $r_2^e$ denote the same for the region B.

overlap of this category of iQTLs (13%) with H3K27ac hQTLs, in comparison to 28% overlap for all iQTLs, while having similar overlap with regulatory regions and TF binding motifs, supports this possibility.

## iQTLs may be associated with concordant changes in chromatin contacts over a broad genomic locus

As previous QTL studies[7,8] report eQTLs regulating expression of multiple genes, we similarly obtained 2090 iQTLs (~22%) associated with multiple HiChIP loops (Supplementary Data 4) potentially involving multiple genes or regulatory regions. We thus enquired whether there exist iQTLs associated with not only one loop but rather with multiple HiChIP loops in a broad locus, and devised a method to infer such QTLs (Fig. 5). Considering an example, the iQTL rs2247325 in the *RNASET2* locus is associated with two HiChIP loops (Fig. 6A, B) while the SNPs rs11980001 and rs6464142 are associated with 6 and 5 HiChIP loops, respectively, in the *NUB1* locus (Supplementary Fig. 9A, B, D). Multiple HiChIP loops associated with a common iQTL could share one common anchor (+/− 5 kb) but may span different genes or regulatory elements over a broad region with their other anchors. In view of this, we next enquired whether any iQTL $X$ is associated with changes in chromatin contacts over a broad region $R$, and whether such changes are accompanied by changes in gene expression within that region. For an iQTL $X$ and its associated HiChIP loop $L$, we define such a testable

region $R$ by adapting the connected component labeling algorithm (see ref. 89 and our previous implementation of merge filtering in FitHiChIP[64]). We utilized the spatial adjacency of the interacting bins of HiChIP loops (see Methods) to define the connected component and underlying HiChIP loops, and defined the testable region $R$ using the union of interacting bins (denoted by the segments A and B in Fig. 5) of these HiChIP loops. If the segments A and B overlapped, the region $R$ was defined by the complete span (A+B) of these regions, while for distinct segments, $R$ was defined by disjoint intervals A and B. An iQTL $X$ was defined as a *connectivity-QTL* for region $R$ provided the complete set $S$ of HiChIP loops (either within A+B or between the disjoint A and B regions) and the corresponding contact map showed significant variation associated with different genotypes of $X$ (see Methods and Fig. 5). Overall, we obtained a set of 218 connectivity-QTLs across 29 loci (regions) (linear-regression $p$-value $<10^{-5}$, Bonferroni adjusted $p$-value $<0.005$, and $>0.1$ connectivity difference between genotype aggregated contact maps; see Methods), where underlying HiChIP loops spanned from distances of up to 1.55 Mb (Supplementary Data 10). Majority (182 out of 218, or ~83%) of these connectivity-QTLs were eQTLs of nCD4, while only 26 iQTLs (~12%) were not eQTLs in any CD4 T cell subsets (Supplementary Data 4, 10) indicating that the changes of chromatin conformation over a broad region is highly likely to be accompanied by changes in gene expression. This was also

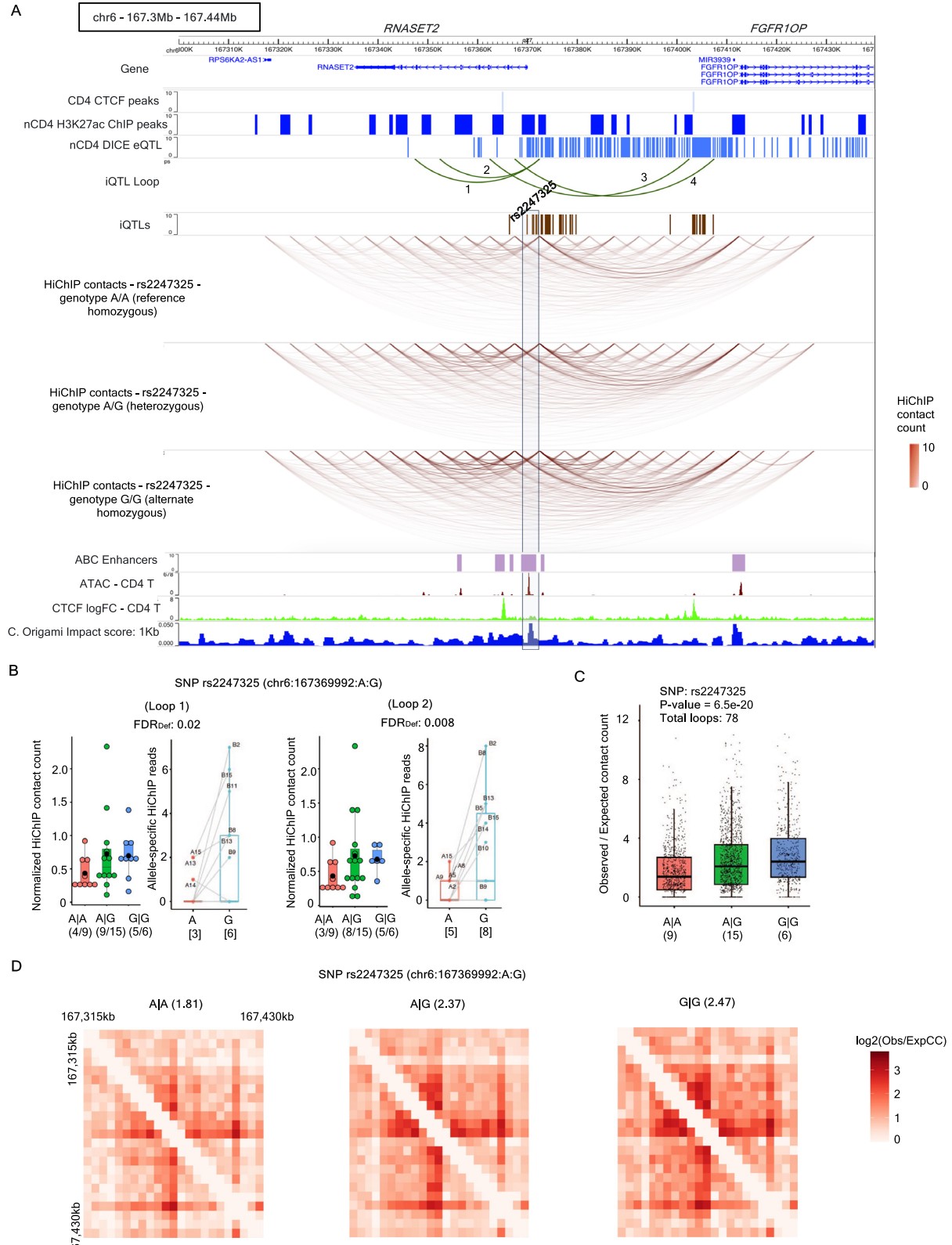

consistent with the high concordance between the direction of eQTL effects and connectivity-QTL effects with 99% (180/182) of the cases the genotype corresponding to higher expression also associated with higher connectivity.

Considering the 115kb *RNASET2* locus that harbors SNPs associated with multiple autoimmune disease, we identified 4 HiChIP loops associated with 51 iQTLs (Supplementary Data 4), out of which the iQTL rs2247325 was found to be a connectivity-QTL (Fig. 6A, Supplementary Data 10). This SNP is also a strong eQTL of *RNASET2* for various CD4 T cell subsets (Supplementary Fig. 8A) and an iQTL for two HiChIP loops both with significant trends for genotype-dependent and allele-specific contact counts (Fig. 6B). As a connectivity-QTL, rs2247325 shows a significant genotype-dependent variation of contact counts for the associated HiChIP loops (Fig. 6C) and for overall

**Fig. 6 | Example of connectivity-QTLs. A** Example of a connectivity-QTL rs2247325 associated with the connectivity of *RNASET2* locus containing overall 4 iQTL-associated loops (numbered), two of which (numbered 1 and 2, shown in the track iQTL Loop) are associated with the iQTL rs2247325. Three tracks below iQTLs indicate the changes in overall HiChIP connectivity (color scales indicate the contact counts) according to the genotypes of rs2247325. ABC model enhancers, corresponding to the E2G links of the activated naïve CD4 T cell type, are also shown. Bottom three tracks show CD4 T cell ATAC-seq, CTCF log fold change over control ChIP-seq track (used as an input to C. Origami) and the impact score at 1kb resolution obtained by C. Origami, where higher impact score indicates higher impact on 3D chromatin organization by perturbation of the corresponding loci. **B** Genotype-dependent variation of sequencing-depth normalized HiChIP contacts and allele-specific variation of HiChIP reads for the iQTL rs2247325 and for the loops 1 (left) and 2 (right). $FDR_{Def}$ indicates statistical significance (FDR) of these SNP-loop pairs using the default RASQUAL model. In the genotype-dependent plots, numbers in the format (***a*** / ***b***) denote that the corresponding genotype is present in ***b*** donors, out of which ***a*** donors have this HiChIP loop as significant (by FitHiChIP). In the allele-specific plots, X axis denotes different alleles, and the underlying numbers indicate the number of donors having nonzero HiChIP read counts for the respective alleles. For genotype-specific plots, *p*-values are obtained from linear regression (ANOVA) while for allele-specific plots, *p*-values are computed by one-sided paired *t*-test. Boxplots show mean (bigger black dots), median (middle lines), 25th, 75th percentiles (box) and individual samples (smaller black dots or dots with symbols A* where * indicates numbers). **C** Genotype-dependent variations of the observed vs expected contact counts across all HiChIP loops within this locus (shown in **A**). Counts are stratified with respect to the SNP rs2247325. Here, X axis denotes different SNP genotypes, and the underlying numbers indicate the number of donors having that genotype. The *p*-values are obtained from linear regression (ANOVA). Boxplot shows median (middle lines), 25th, 75th percentiles (box) and individual samples (black dots). **D** Aggregated contact map (log2-transformed observed vs expected contact counts) for different genotypes of the SNP rs2247325. The numbers on top are the 95th percentile values of the respective contact maps. Obs: observed, Exp: expected, CC: contact count. Source data are provided as a Source Data file.

connectivity (all HiChIP contact counts) of this region with both effects in the same direction as the *RNASET2* expression (Fig. 6D, Supplementary Fig. 8A). However, genotype or allele-specific variation of 1D ChIP and HiChIP coverages for this SNP are not significant suggesting that this connectivity-QTL directly affects the 3D genomic chromatin contacts (Supplementary Fig. 8B). Interestingly, the SNP rs2247325 is in only moderate LD ($R^2 \sim 0.65$) with the rest of the 50 iQTLs that are tightly linked ($R^2 > 0.8$) (Supplementary Fig. 8C). The SNP rs2247325 is a regulatory SNP and is also linked to the GWAS studies for various diseases such as Crohn's disease, various autoimmune traits, hypothyroidism, diabetes, inflammatory bowel disease (IBD), etc. according to Open Targets Genetics[80,81] and PheWAS (https://a2f.hugeamp.org/) analysis. Genome-wide perturbation screen predicted by the deep learning method C. Origami[86] using the training model defined by reference CD4 T cell Hi-C data (provided in ref. [59]) shows that the connectivity-QTL rs2247325 overlaps with loci with the highest impact score (Fig. 6A). The SNP also overlaps with the CD4 T cell ATAC-seq peak, and an activated CD4 Naïve cell ABC enhancer with an enhancer-to-gene link to *RNASET2*[66]. Strong eQTL trend accompanied by substantial changes in chromatin contacts within this locus suggests a concerted genotype-dependent variation in regulation of *RNASET2* consistent with the identification of this gene in many GWAS and post-GWAS functional analysis of immune cells.

Another 6 connectivity-QTLs, including the previously mentioned SNPs rs11980001 and rs6464142, belong to a 200kb region involving *NUB1* gene (Supplementary Fig. 9A) and show a significant genotype-dependent variation of HiChIP contact counts and aggregated contact maps (Supplementary Figs. 9B-G). All 6 connectivity-QTLs are in strong ($R^2 > 0.8$) LD (Supplementary Fig. 9H), and are eQTLs of nCD4 (ImmuNexUT) for the genes *CRYGN* and *WDR86*, however with both genes having very low expression (<3 TPM on average). The iQTL rs11980001 is of particular importance (Supplementary Fig. 9A) since it is also an eQTL of the interacting gene *NUB1* in naïve CD8 (ImmuNexUT) (Supplementary Fig. 9I), overlaps with nCD4 H3K27ac ChIP-seq peaks, and the genotype-dependent variation of HiChIP contact counts matches with the expression trend of *NUB1* in naive CD4 DICE data (Supplementary Fig. 9J). The locus containing these SNPs also exhibits high impact scores from C. Origami and overlaps with CTCF log fold change tracks, thereby may be a potential regulator of 3D chromatin organization. This locus also overlaps with the ABC enhancers of activated naïve CD4 T cell having E2G links to both *CRYGN* and *WDR86* (Supplementary Fig. 9A). Open Targets Genetics also confirms PCHi-C interactions between rs11980001 and *NUB1* for naïve CD4 and CD8 cell types. Furthermore, this SNP is also identified as a single-cell eQTL for the memory T cells[11], showing its potential importance in regulation of *NUB1* across different T cell subsets.

Both examples so far have highlighted connectivity-QTLs associated with a contiguous region. We then focused on connectivity-iQTLs in the *KANSL1* locus (chr17) where the associated HiChIP loops are between two disjoint regions ~350 kb apart spanning 65kb (**A**) and 210kb (**B**), respectively (Supplementary Fig. 10A). This locus has been previously marked for containing fusion transcripts related to T-ALL leukemia[90]. The *KANSL1* gene is also related to a variety of cellular processes including enhancer regulation, cell proliferation, and mitosis. Overall, we obtained 44 connectivity-QTLs in this locus, all of them in tight ($R^2 > 0.8$) LD, showing significant genotype-dependent changes of HiChIP contact counts and aggregated contact maps (Supplementary Figs. 10B-D). Further, 38 of these 44 iQTLs are eQTLs not only for the nCD4 but for all 15 cell types in the DICE database (Supplementary Data 4, 10). The 17q21.31 locus with these QTLs also harbors a common inversion polymorphism spanning up to 1Mb that includes our A and B regions and all HiChIP loops connecting them. The 17q21.31 inversion polymorphism exhibits two primary haplotypes: non-inverted (H1) and inverted (H2), also known as the MAPT haplotypes. These are connected to diverse diseases including neurodegenerative, developmental, and immune-related disorders. Additionally, they are linked to normal variations in common traits such as brain size. Notably, recent cerebral cortex GWAS pinpointed this inversion as a top hit linked to diminished overall cortical surface area, particularly prominent in the visual cortex[91,92]. One of the connectivity-QTLs we identified, rs62071575 (also named rs554959222), was among the numerous SNPs reported for their high LD with the inversion, where the C allele was noted to tag this inversion ($R^2$ of 0.94)[93]. This SNP was also an eQTL for *KANSL1* gene (within the inverted region) and *ARL17A* gene (at the right boundary of the inversion) with inverted allele showing higher expression for both genes (Supplementary Fig. 10E) consistent with previous reports[91,92]. The genotype-dependent connectivity pattern of the overall region for this SNP revealed strong proximity of the two inversion endpoints especially for donors with C/C genotype confirming the 1D/genomic proximity or juxtaposition of these endpoints upon inversion. Although this is a genetic event rather than an epigenetically driven change in contact maps, this case confirms the validity of our connectivity-QTL mapping in identifying genotype-dependent patterns in chromatin contacts.

## Discussion

Despite robust associations between genetic variants and several human diseases, the molecular mechanisms of how noncoding genetic variants perturb gene expression remain unclear, particularly for variants outside the immediate vicinity of gene boundaries[94]. Although various studies exist to identify genetic variants associated with gene expression (eQTLs) or 1D chromatin accessibility (caQTLs), our study characterizes the genetic variants altering chromatin interactions

among active regulatory regions by integrating both between (genotype-dependent) and within (allele-specific) individual information. In addition, our study characterizes such interaction QTLs in primary cells instead of cell lines[44,58]. Our choice of naïve CD4 T cells was motivated by their relative homogeneity compared to other immune cell types and by the discovery of allele-specific chromatin contacts in the ~300 kb 17q21 asthma locus in our earlier study[77]. Our identified QTL associations between ~9K SNPs and ~2K HiChIP loops after multiple layers of strict filtering constitute the largest set of such associations to date although larger sample sizes will be needed to discover associations potentially missed by this study.

Our choice of using HiChIP to capture protein-centric (here H3K27ac) chromatin contacts was mainly for lower sequencing requirement compared to Hi-C to discover high-resolution loops. However, since HiChIP is a protein-centric 3C approach, the derived iQTLs and the corresponding changes in chromatin contacts may actually be attributed to changes in 1D histone modifications. Using the reference Blueprint H3K27ac hQTLs, however, we found that ~62% iQTLs are not overlapping and are not in LD with hQTLs and only 2.3% of hQTLs are iQTLs suggesting the correspondence between 1D ChIP-seq and 3D HiChIP signals is far from one-to-one. Concordance of effect sizes between shared iQTLs and hQTLs, however, confirmed that the changes in 1D and 3D are generally in the same direction. This was also the case for iQTLs and eQTLs with strong concordance for overall set of iQTLs and nearly 100% concordance for the small set of connectivity-QTLs that are a subset of all iQTLs.

To understand better the extent at which genetic variants explain changes in both chromatin looping and gene expression, our work categorized iQTLs with respect to their overlap with eQTLs in either naïve CD4 T cells or other CD4 T cell subsets. As ~36% iQTLs cannot be explained by eQTLs (neither overlapping nor in high LD), we reasoned that changes in 3D chromatin loops may not always be reflected in the gene expression measured simultaneously. Recent studies on modeling the dynamics of genetic effects on gene expression in either different timepoints of cellular differentiation[15,85], activation[12,13,77], or transcriptional trajectories across single-cell RNA-seq data[95] motivated us to ask whether 3D looping changes due to genetic variants lead to the changes in gene expression later in CD4 T cell differentiation. Indeed, ~12% iQTLs were found to be eQTLs in derived CD4 T cell subsets supporting this hypothesis. We demonstrated this concept using *MICB* locus as an example where the presence of iQTL signal did not translate into eQTL signal in nCD4 but did so in Treg memory and other T helper subsets. Derived iQTLs, whether they overlap with eQTLs or not, were shown to be enriched with fine-mapped GWAS SNPs and carried higher heritability information for various immune diseases, suggesting that chromatin interactions and the underlying regulatory regions may point to potentially causal variants associated with disease risk.

This class of connectivity-QTLs we propose here is conceptually different from the QTLs employing various features derived from the Hi-C contact matrices such as insulation score (INS), directionality index (DI) and frequently interacting region (FIRE) scores[58]. This is because connectivity-QTLs are defined using a set of loops and a slice of the contact matrix rather than features collapsed into 1D space. Due to exponential number of combinations, it will be prohibitive to do a de novo connectivity-QTL mapping for all SNPs and combinations of loops in their vicinity. Therefore, using iQTLs as a starting point and using an algorithm to define connected components made it possible for us to define statistically significant associations of genetic variants with chromatin contacts within a predefined region. Majority of these connectivity-QTLs were also found to be nCD4 eQTLs with concordant directionality, suggesting that concerted changes in 3D chromatin organization are usually accompanied by changes in gene expression.

The increasing use of deep learning techniques to model 3D chromatin interactome using DNA sequence and epigenomic tracks motivated us to assess the iQTLs, connectivity-QTLs and corresponding loci, using a recently published deep learning method C. Origami[86]. The published CD4 T cell Hi-C data from[59] enabled us to create a trained model on CD4 T cells and apply in silico genetic screening on the selected loci to assess their regulatory impacts at 1 kb resolution. These screens highlighted partial overlap of iQTLs with regions predicted to have high impact on the 3D genome organization when deleted. In addition, we used the recent popular approach of linking enhancers to target genes by activity-by-contact model[19,55] using their latest predictions of enhancer-to-gene (E2G) links[66]. These comparisons showed a substantial fraction of iQTLs were overlapping or within the vicinity of predicted ABC enhancers. Given the non-trivial nature of validations required for iQTLs due to linkage disequilibrium among SNPs and overall connectivity of the analyzed regions beyond a specific loop, such in silico predictions provide valuable orthogonal evidence about the importance of iQTLs.

The developed iQTL and connectivity-QTL discovery framework can be applied to other 3C techniques such as Hi-C, Micro-C, PCHi-C and ChIA-PET, with modifications in data pre-processing, normalization and loop calling. Similarly, QTL discovery methods other than RASQUAL can also be incorporated as appropriate. One possible extension of the proposed iQTL framework would be to incorporate the fine-mapping technique on the RASQUAL iQTL summary statistics for individual SNP-loop pairs, by adapting the procedure for 1D-QTLs employed in PLASMA[96]. This would alleviate the issue of LD between iQTLs and may further prioritize iQTLs that are causal. However, development and validation of such a fine-mapping approach is a challenging computational problem that can be a subject of future work.

The statistical power of detecting these interaction QTLs depends on the sample size and the sequencing depths of 3C samples. In future, 3C datasets with higher sample sizes and sequencing depth would be useful to assess the robustness of iQTLs according to varying input conditions. Given iQTLs from multiple cell types, future works may also involve linking cell-specific iQTLs and cell-specific chromatin loops (for different TFs or histone marks) to understand the genetic drivers behind cell-specific gene regulatory mechanisms.

# Methods

### Leukapheresis samples
The DICE study was approved by the Institutional Review Board (IRB) of the La Jolla Institute for Immunology (LJI; IRB protocol no. SGE-121-0714). This study involved a total of 91 healthy volunteers who were recruited in the San Diego area and signed written, informed consent for collection of leukapheresis samples (at the San Diego Blood Bank) and for sharing deidentified data for research purposes. All study subjects self-reported ethnicity and race details, and were tested negative for hepatitis B, hepatitis C, and human immunodeficiency virus (HIV). The iQTL study reported here used samples from 30 donors selected among the 91 DICE donors for which the naïve CD4+ T cells were sorted previously as part of an earlier publication[8] with gating strategy described in Fig. S1 of that work.

### HiChIP data generation
We employed the HiChIP protocol described in our earlier work[47] to generate the CD4 Naive T cell H3K27ac HiChIP data (Supplementary Data 1). We also used CD4 Naive T cell H3K27ac HiChIP data of 6 donors (dbGaP accession number *phs001703.v3.p1*) published in our previous study[47].

### HiChIP data processing and loop calling
We processed CD4 Naïve T cell H3K27ac HiChIP data of 30 donors using the procedure described in our earlier work[47]. Briefly, HiChIP paired-end fastq reads were aligned by HiC-Pro[97] (version 2.9.0) using hg19 as the reference genome. In HiC-pro, we used bowtie2[98] for

alignment and used the bowtie2 global options *--very-sensitive -L 30 --score-min L,-0.6,-0.2 --end-to-end --reorder* and the local options *--very-sensitive -L 20 --score-min L,-0.6,-0.2 --end-to-end --reorder* in the HiC-pro configuration file. We used MAPQ threshold = 0 and MboI as the restriction enzyme for read mapping. Aligned reads were then paired, and paired reads involving two different MboI restriction sites were retained. Duplicate reads were removed using Picard (https://broadinstitute.github.io/picard/). We used FitHiChIP[64] (version 9.1) to call the significant HiChIP loops from individual donors, using the ChIP-seq peaks obtained across all DICE donors (Supplementary Data 2 - unpublished data). We used the following parameters in FitHiChIP for loop calling: 5kb resolution (*BINSIZE*=5000), peak-to-all output inter-actions (*IntType*=3), distance range 10kb - 3mb, loose (L) background (*UseP2PBackgrnd*=0), coverage bias regression (*BiasType*=1), no merge filtering (*MergeInt*=0), and 1% FDR (*QVALUE*=0.01). Union of significant loops from all donors involving only the autosomal chromosomes were used for iQTL analysis. FitHiChIP interactions and the iQTLs were all visualized using WashU epigenome browser[99–101]. We also called peaks from HiChIP data using the *PeakInferHiChIP.sh* utility function of FitHiChIP.

### Genotyping and imputation of SNPs

We used the SNP genotype data previously published in the DICE database[8]. SNP calls from the Infinium Multi-Ethnic Global-8 Kit (Illumina) were exported using GenomeStudio v2011.1 (Illumina). The data quality was assessed using snpQC[102] and low-quality SNPs (SNPs failing in more than 5% of the samples and SNPs with Illumina's GC scores less than 0.2 in more than 10% of the samples) were removed for down-stream analysis. Sex, ethnicity and relatedness of the subjects was inferred from the genotype data using PLINK v1.90b3w[103]. IMPUTE[104] (version 2.3.2) was used to phase observed genotypes and impute SNPs using the 1000 Genomes Project[105] phase 3 as the reference panel, resulting in >80 million imputed SNPs for autosomal chromosomes.

### Principal component analysis (PCA) on HiChIP donors

We applied PCA on the FitHiChIP loops of 30 donors using the procedure described in our earlier work[47]. Briefly, we extracted top 10,000 significant (FDR < 0.01) FitHiChIP-L loops (loose or P2P=0 background) from individual donors, resulting in a union of 37,180 loops. We created a feature matrix $M$ of dimension 37180 X 30, containing the raw HiChIP contact counts of individual donors. Missing values in $M$ were replaced by 0. The matrix $M$ was then applied to the R function *prcomp* to compute PCA. Individual samples in the PCA were plotted using the R package *factoextra* and using the covariates like sex, age group or race.

### Identifying iQTLs (HiChIP 2D QTLs)

We adapted the RASQUAL[65] pipeline to derive the significant iQTLs and corresponding HiChIP loops. RASQUAL is widely used to identify the eQTLs using RNA-seq based gene expression counts and chromatin-QTLs using either ChIP-seq chromatin signal or ATAC-seq chromatin accessibility data. RASQUAL supports modeling both genotype dependent deviation of the observed gene expression (or chromatin accessibility) and the allele-specific changes of the gene expression (or chromatin accessibility). Although our previously published DICE[8] eQTL database used MatrixQTL[106] to infer the eQTLs, here we used RASQUAL for iQTL analysis since it incorporates both allele-specific variation of the HiChIP reads and the genotype dependent trend of the HiChIP contacts, while MatrixQTL only models the later without processing allele-specific reads.

### Proxy of gene expression

To apply RASQUAL for HiChIP interactions, we used the union of FitHiChIP-L (peak-to-all interactions using loose or P2P=0 background) significant loops for all donors. HiChIP contact counts of these loops for individual donors were used as a proxy of the donor-wise gene expression.

### SNPs considered

Conventional eQTL analysis using MatrixQTL[106] considers all SNPs within 1 Mb from the transcription start site (TSS) of a target gene, and examines the statistical significance of all such SNP-gene pairs. In the current iQTL analysis, for each loop between two 5kb interacting bins $b_1$ and $b_2$, we considered only the SNPs within a pre-defined window ($w$) of these interacting bins, in particular, SNPs within the intervals $[b_1 - w, b_1 + w]$ and $[b_2 - w, b_2 + w]$. Given the HiChIP loops in the current study employed 5kb resolution, we used $w = 5000$ (5kb, similar to the loop resolution), thus effectively considered the SNPs falling within two 15kb intervals. We considered only the bi-allelic SNPs in autosomal chromosomes for iQTL analysis.

### Estimating allele-specific reads

For RNA-seq and ChIP-seq data, allele-specific reads are estimated using their respective alignment (BAM) files, by computing the number of reads overlapping with a specific allele. HiChIP contact counts between interacting fragments, however, depend on the binding of regulatory proteins (here H3K27ac), efficiency of proximity ligation, restriction sites, and the genomic distance between these fragments. Thus, estimation of allele-specific reads needs to account for the variation of contact counts according to genomic distance, mappability, and restriction site positions. In view of this, for a given SNP, we did not use a fixed pair of allele-specific read counts, but rather estimated the allele-specific reads separately for each HiChIP contacts whose interacting bins overlapped with that SNP. We only used paired-end *cis* contacts within 10kb – 3Mb to estimate the allele-specific reads. For a particular HiChIP contact between two 5kb bins $b_1$ and $b_2$, allele-specific reads of a SNP falling in either of the regions $b_1 +/- w$ or $b_2 +/- w$ (here $w$ = 5kb) were computed donor-wise by counting the number of paired-end HiChIP reads (for a specific donor) from that SNP to the opposite 15kb interval. Specifically, for a SNP within the region ($b_1 +/- w$), its allele-specific reads were computed by counting the number of reads between the SNP (with specific allele configuration) and the region ($b_2 +/- w$). The *ASEReadCounter* routine of the toolkit GATK[107] (https://gatk.broadinstitute.org/hc/en-us) with the option *--min-mapping-quality 10* was used to estimate the allele-specific reads.

### Covariates

In addition to the default set of covariates estimated by RASQUAL, we used the sequencing depth of individual HiChIP data and also the sex, ethnicity, race and age information of individual donors (available in DICE[8] database) as the covariates. We applied RASQUAL scripts *makeCovariates.R* and *makeOffset.R* on the complete data from all autosomal chromosomes to compute the covariates and offsets. We then used the *txt2bin.R* script in RASQUAL to convert the covariates, offsets, and coverage values into binary format supported by RASQUAL.

### iQTL models

We executed RASQUAL on the input set of SNPs and loops separately for individual chromosomes, using three different settings:

1. *Default* model: considers both genotype-dependent variation of HiChIP contact counts and the variation of allele-specific reads from heterozygous donors. We have used the symbol $FDR_{Def}$ to denote the significance values generated from this model.
2. *Genotype dependent* model (RASQUAL command line option *--population-only*): uses only the genotype-dependent variation of HiChIP contact counts but does not consider the allele-specific variation of HiChIP reads. We have used the symbol $FDR_{Pop}$ to denote the significance values from this model.
3. *Allele-specific* model (RASQUAL command line option *--as-only*): uses only the allele-specific variation of HiChIP reads from

heterozygous donors to compute the statistical significance values which we denote by the symbol $FDR_{AS}$.

In addition to RASQUAL's allele-specific model, we have also performed a paired *t*-test between the allele-specific reads for individual heterozygous SNPs, and denoted the resulting *p*-values by the symbol $P_{AS}$. Reason for employing three different models of RASQUAL was to assess the contribution of both genotype dependent and allele-specific variations of HiChIP contacts and reads. Initially we selected the iQTL - loop pairs which were significant by the default model ($FDR_{Def} < 0.05$). However, lots of entries significant by the default model were not significant in either genotype dependent or allele-specific models of RASQUAL. In view of this, to eliminate the potential false positive entries, we selected only those SNP - loop pairs which satisfy all of the following conditions:

1. SNPs should have all three genotypes (reference homozygous or 0|0, heterozygous or 0|1, and alternate homozygous or 1|1) present in at least two donors.

2. If the current SNP-loop pair is not significant by the genotype dependent model ($FDR_{Pop} >= 0.05$), allele-specific reads should produce statistical significance by the paired *t*-test analysis ($P_{AS} < 0.05$). Note that we did not consider the statistical significance in allele-specific model of RASQUAL ($FDR_{AS}$) because we observed that the allele-specific model of RASQUAL often returns false positive entries without any specific trends of the allele-specific reads and these trends get rejected by the paired *t*-test. In view of this, we used the paired *t*-test as a criterion to select the SNP-loop pairs significant by the allele-specific reads, even if they are not declared significant by the RASQUAL's allele-specific model.

3. If the SNP-loop pair is significant by the genotype dependent model ($FDR_{Pop} < 0.05$) and the allele-specific read trend is not significant by the paired *t*-test ($P_{AS} >= 0.05$), either mean or median of the raw and normalized HiChIP contact counts (normalized by sequencing depths) per genotype should exhibit either strictly ascending or strictly descending trends between three different genotypes, namely reference homozygous, heterozygous and alternate homozygous. Otherwise, the entry would be filtered out.

4. If the SNP-loop pair is significant only by the paired *t*-test of allele-specific reads ($FDR_{Pop} >= 0.05$ and $P_{AS} < 0.05$), allele-specific reads need to follow the same trend as the genotype dependent HiChIP contacts. For example, considering a SNP with two alleles are A and G such that the number of reads for A are higher than those for G, the normalized HiChIP counts for the genotype A|A should also be higher than the genotype G|G. Otherwise, those SNP-loop pairs would be filtered out.

5. Finally, the SNP-loop pairs significant in the default RASQUAL model and satisfying all of the conditions 1 to 4 are retained.

**Aggregate peak analysis (APA) of HiChIP and Hi-C loops**
We performed APA using the R package GENOVA[108]. For the HiChIP data, the background set of HiChIP contacts were derived by merging all the valid read pairs (generated from HiC-pro) from all input 30 donors and applying ICE[109] normalization on the merged HiChIP contacts using the FitHiChIP[64] pipeline. HiChIP loops associated with iQTLs were compared against the union of significant FitHiChIP-L loops from all donors. For a candidate loop, we used the background HiChIP contacts 50kb up- and downstream, thus generating a matrix of $21 \times 21$ dimension corresponding to 5kb resolution (as suggested in ref. 110). Loops within 150kb – 1mb distance were only considered. For the Hi-C data, we downloaded the hg38 merged CD4 T cell Hi-C data provided in ref. 59 from the link http://bartzabel.ls.manchester.ac.uk/orozcolab/SNP2Mechanism/hic/merged/, performed KR normalization and used as the background Hi-C contacts.

**Pathway analysis**
We used the protein-coding genes having expression > 1 TPM in CD4 Naive cell type (according to the DICE database[8]) and performed exact overlap with the iQTL loop anchors (5kb bins) to derive the candidate genes for pathway analysis conducted using Metascape[11]. Results of gene ontology analysis with respect to the biological processes were reported.

**Analysis of published eQTLs for CD4 Naïve and other CD4 T cell subsets**
We downloaded eQTLs for CD4 Naïve and various other CD4 T cell subsets (CD4 Stimulated, Tfh, Th1, Th2, Th17, Th1/17, Treg Memory and Treg Naïve) from the DICE[8] database (https://dice-database.org). We also downloaded the conditional eQTLs (FDR < 0.05) for the CD4 Naive and other CD4 T cell types (Memory CD4, Tfh, Th17, Th1, Th2, Fr-I-nTreg, Fr-II-eTreg and Fr-III-T) from the ImmuNexUT[7] database (https://www.immunexut.org/) and used the conditional eQTLs having the rank of association ($10^{th}$ field) as 0.

**Overlap of QTLs with reference regulatory regions (peaks) for various CD4 T Cell subsets**
We used the following three sets of reference peaks to check their overlap with the iQTLs or reference eQTLs.

1. CD4 Naïve H3K27ac ChIP-seq peaks: We utilized the aggregate ChIP-seq peaks file obtained across all DICE donors (unpublished data).

2. CD4 Naïve H3K27ac HiChIP peaks: We merged the valid paired-end HiChIP *cis* reads within the distance range 10kb – 3mb from all iQTL donors, and applied this merged set of reads to the utility routine *PeakInferHiChIP.sh* in FitHiChIP[64] to call the HiChIP peaks.

3. ChIP-seq peaks for various CD4 T cell subsets: We downloaded the ChIP-seq peaks for various CD4 T cell subsets and transcription factors (reference genome: hg19) from the ChIP-Atlas[67] database (https://chip-atlas.org/), using the peaks with q-value < 1e-5 (threshold of significance = 50).

4. ChIP-seq peaks from ReMap2022 database: We also downloaded ChIP-seq peaks from ReMap DNA binding database[88]. Specifically, we considered the ChIP-seq peaks marked for CD4 T cells.

We used the exact overlap criterion to identify the QTLs (iQTLs or eQTLs) falling within any of these peaks.

**Overlap of iQTLs with reference transcription factor binding sites and motifs**
To identify the TF binding motifs overlapping iQTLs, we employed multiple approaches, as described below:

1. ADASTRA database: We downloaded the Bill Cypher version (5.1.2) of the ADASTRA database[71,72] (https://adastra.autosome.org/) having information of various human TF binding sites for different cell lines and cell types, and overlapped these binding sites with the iQTLs.

2. De novo TF motif enrichment by FIMO: For each chromosome, we first extracted +/− 20 bp around individual iQTLs (thus extracted 41 bp sequences) and created two different fasta files such that the first fasta file contains the reference alleles of the input iQTLs while the second file contains the alternate alleles. Individual fasta files per chromosome were then applied to FIMO[69]. We used three different motif databases and corresponding reference motifs: 1) cisbp[112] (version 1.02, file *Homo_sapiens.meme*), 2) hocomoco[113] (file *HOCOMOCOv11_core_HUMAN_mono_meme_format.meme*) and 3) jaspar[114] (file *JASPAR2018_CORE_vertebrates_non-redundant.meme*), to identify the motifs. Motifs with *p*-value < 1e-5 were considered significant. Statistically significant motifs in at least one database were reported. To plot the motifs, genes with expression > 1 TPM in the CD4 Naïve cell type (according to the

DICE[8] database) and the most significant motifs (having $p$-value < 1e-6) were only considered. Similar procedure was employed to compute the TF motif enrichment of CD4 Naïve eQTLs in the DICE database. Specifically, we used the DICE eQTLs for autosomal chromosomes and considered motifs with $p$-value < 1e-5 as significant. Fisher's exact test was used to compute the enrichment ($p$-value) of TF binding motif overlap between iQTLs and DICE CD4 Naïve eQTLs.

3. Allele-specific TF binding motifs by AME: Allele-specific motif enrichment analysis was performed by the procedure outlined in various reference studies[51,115]. We used AME[70] from the MEME suite[68], and used position weight matrices (PWM) from the above mentioned three motif databases (cisbp, hocomoco and JASPAR) to predict the motif enrichment. We extracted 41 bp sequences (+/− 20 bp) around individual iQTLs (SNPs) for all the autosomal chromosomes together. We used sequences from the effect alleles (alternate alleles) as the test set, and used the shuffled background as the control sequences. Genes with expression > 1 TPM in the CD4 Naïve cell type (according to the DICE database) and motifs with $p$-value < 1e-6 were only plotted.

### Overlap and LD with fine-mapped GWAS SNPs

We downloaded the fine-mapped GWAS SNPs from the CausalDB[73] database (http://www.mulinlab.org/causaldb/). Specifically, we downloaded the file *credible_set.20220315.tar.gz* and selected only the GWAS studies involving various immune diseases. The SNPs reported in the credible sets of corresponding studies were overlapped with the iQTLs and DICE CD4 Naive eQTLs using the fields like rsID, chromosome and SNP position. We used Fisher's exact test to compute the enrichment ($p$-value) of overlap between iQTLs and reference DICE CD4 Naive eQTLs. Linkage Disequilibrium (LD) analysis was carried out using the tools LDmatrix and LDpair of the LDlink web repository (https://ldlink.nci.nih.gov/) and using the European population as the reference.

### Significance of various overlap analyses

**Overlap of iQTLs with reference CD4 Naïve T cell eQTLs and Blueprint Histone QTLs.** As discussed in Section A, ~64% (5993 out of 9426) iQTLs overlap with either CD4 Naïve T cell eQTLs or Blueprint histone QTLs. To create a null distribution of SNPs, we checked each iQTL associated HiChIP loops (total 2292 loops) and for each loop, sampled the same number of SNPs as the number of iQTLs associated with this loop. For each loop, we considered only the SNPs either overlapping or within +/− 20kb of the loop anchors, for sampling (without replacement) such that the sampled SNPs did not overlap with the list of iQTLs. Finally, the sampled 9426 SNPs (same number of SNPs as the number of iQTLs) were tested for overlap with the reference CD4 Naïve T cell eQTLs in either DICE or ImmuNexUT databases, or the list of BLUEPRINT histone QTLs. The 2X2 contingency table for the SNPs overlapping with either eQTLs or hQTLs is reported in Supplementary Data 11A. The same procedure is applied to compute the enrichment of iQTLs overlapping with CD4 Naïve T cell eQTLs (Supplementary Data 11B).

**Overlap of loop anchors with reference CD4 Naïve T cell eQTLs and Blueprint Histone QTLs.** To benchmark the overlap of iQTL associated HiChIP loop anchors with the reference CD4 Naïve eQTLs and BLUEPRINT histone QTLs, we considered the 2292 HiChIP loops associated with the iQTLs, and extracted these 5kb loop anchors and surrounding +/− 5kb intervals to finally come up with a unique list of 8060 intervals of 5kb resolution. To get the control set of FitHiChIP loop anchors, we first extracted top 20000 FitHiChIP loops (approximately 10 times the number of iQTL associated HiChIP loops) according to the number of donors showing statistical significance (FDR < 0.01) for these loops, and extracted the loop anchors exclusive to the previously

constructed iQTL associated loop anchors. In process, we obtained 12719 FitHiChIP loop anchors. We then randomly sampled the same number (12719) of 5kb bins (generated using the command *bedtools makewindows*) exclusive to both iQTL and FitHiChIP loop anchors from the complete set of chromosomes, to generate the control of randomly sampled 5kb bins. For individual sets of 5kb bins, we then checked their overlap with the reference CD4 Naïve DICE eQTLs and the BLUEPRINT histone QTLs, and reported corresponding numbers in Supplementary Data 11C.

**Overlap of ChIP-seq peaks with reference CD4 Naïve T cell eQTLs and Blueprint Histone QTLs.** We considered the ChIP-seq peaks used in the current study for FitHiChIP loop calling, and distributed into two sets: 1) peaks falling into the above mentioned iQTL associated HiChIP loop anchors (5kb regions) and 2) the remaining set of peaks. For both sets of peaks, we carried out the overlap analysis as described above, and reported the contingency table in Supplementary Data 11D.

### Stratified LD score regression (S-LDSC) analysis

The stratified LD score regression[74,75] (S-LDSC) technique estimates partitioned heritability from GWAS summary statistics, and reports the enrichment of heritability for a specific set of regulatory regions, such as a candidate set of ChIP-seq peaks. The heritability enrichment[74,75] is defined as the ratio of the proportion of heritability explained by the set of input SNPs, to the proportion of input SNPs compared to the total number of SNPs. To apply S-LDSC to find the heritability enrichment of a given set of SNPs such as the iQTLs and the DICE CD4 Naïve eQTLs, we adapted the S-LDSC method in a custom script hosted in https://github.com/ay-lab/S_LDSC_SNP, where, instead of the ChIP-seq peaks (or regulatory regions), we assessed the heritability enrichment of a given set of SNPs. We downloaded reference GWAS summary statistics for various immune diseases from the GWAS catalog[79] and followed the LD score regression tutorial in GitHub (https://github.com/bulik/ldsc/wiki). We used the LD scores and PLINK[103] generated files from the 1000G Phase 3 release datasets (files *1000G_Phase3_baselineLD_v2.1_ldscores.tgz, 1000G_Phase3_plinkfiles.tgz, w_hm3.snplist, 1000G_Phase3_weights_hm3_no_MHC.tgz, list.txt, 1000G_Phase3_frq.tgz*) provided in https://alkesgroup.broadinstitute.org/LDSCORE/. The script *munge_sumstats.py* from LDSC Github repository was used to convert the input GWAS summary statistics to the *sumstats* file format compatible with S-LDSC. Conventional analysis in S-LDSC employs the script *make_annot.py* with parameters *--windowsize 1000 --bimfile 1000G.EUR.QC.\*.bim* to extract the SNPs within 1kb window from the input set of peaks, and writes them into LDSC compatible "thin annot" file format. In the current study, as we are considering the SNPs instead of peaks for enrichment analysis, we have instead dumped the specific SNPs (iQTLs or eQTLs) and corresponding annotations from the reference *.bim* files *1000G.EUR.QC.\*.bim* and exported into S-LDSC compatible "thin annot" format. These annotation files were subsequently applied to the script *ldsc.py* with parameters *--l2 --thin-annot --ld-wind-cm 1 --bfile 1000G.EUR.QC.\*.bim* to generate the LD scores. Finally, these LD scores along with the reference GWAS summary statistics were applied on the script *ldsc.py* with parameters *--h2 --w-ld-chr weights.hm3_noMHC. --overlap-annot --frqfile-chr 1000G_Phase3_frq/1000G.EUR.QC.* and conditioned on the 1000G Phase3 baseline LD (version 2.1) scores (provided in https://alkesgroup.broadinstitute.org/LDSCORE/), according to the procedure described in https://github.com/bulik/ldsc/wiki/Partitioned-Heritability to derive the heritability enrichment of the given set of SNPs with respect to the corresponding GWAS summary statistics. These enrichment values together with the significance ($p$-values) were plotted using a custom R script.

## Overlap with reference single cell eQTL databases

We downloaded the following set of eQTLs or variants from various reference studies in order to find their overlap with our derived set of iQTLs.

1. Treg eQTLs in ref. 9: We downloaded 3301 Treg eQTLs from the Supplementary Table 2 of the Bossini-Castillo, L. et al. manuscript[9], where the assay is mentioned RNA. This is because, the database contains a mixture of eQTL and chromatin QTL. As the eQTLs follow hg38 coordinates, we extracted the rsIDs using custom routines to compare their overlap with iQTLs.
2. MPRA prioritized variants (emVars) in ref. 10: We downloaded 313 MPRA validated functional variants (emVars) subject to FDR < 0.1 from the Supplementary Table 3 of the Mouri, K. et al. manuscript[10]. We also downloaded the PICS fine mapping SNPs (95% credible causal set) from the supplementary table 9 of the Mouri, K. et al. manuscript[10].
3. Single cell eQTLs for memory T cells[11]: We downloaded the single cell eQTLs (FDR < 0.05) from the supplementary table 1 of the Nathan, A. et al. manuscript[11]. As the eQTLs follow hg38 coordinates, we extracted the rsIDs using custom routines to compare their overlap with iQTLs.
4. Single cell dynamic eQTLs for naïve T cells[13]: We downloaded the single cell dynamic eQTLs (FDR < 0.05) corresponding to the T_naive cell type from the supplementary table 8 of the Soskic, B. et al. manuscript[13].
5. Single cell eQTLs for CD4 T cell subsets[15]: We downloaded the single cell eQTLs (FDR < 0.05) for various CD4 T cell subsets from the supplementary table 10 of the Yazar, S. et al. manuscript[15].
6. Single cell CD4 T eQTL from sceQTLGen consortium[14]: We downloaded the CD4 T cell eQTLs from sceQTLGen (https://www.eqtlgen.org/sc/datasets/vanderwijst2018.html) and extracted the significant eQTLs (FDR < 0.05) for CD4 T cells.
7. Single cell eQTLs from DICE database[12]: We used the single-cell eQTLs for various CD4 T cell subsets published in the DICE database (https://dice-database.org/).

## Colocalization analysis

We performed colocalization between eQTL and reference GWAS SNPs as described in our previous study[50]. We employed the COLOC framework[24] in our custom implementation of colocalization (https://github.com/ay-lab/Colocalization). We used significant GWAS variants with association $p$-value < $5 \times 10^{-8}$ for colocalization. For all eGenes of DICE CD4 Naive eQTL data, we used the eQTL summary statistics, specifically the association $p$-values, effect size ($\beta$), standard error of $\beta$, and the minor allele frequency (MAF) for all variants within 1 Mb of TSS of these eGenes. We excluded all variants within the MHC locus (chr6: 28,477,897 - 33,448,354). For each eGene, both eQTL and GWAS summary statistics were then applied to the *coloc.abf* routine of the COLOC package. We used the default setting ($p_1$ or $p_2 = 1 \times 10^{-4}$) for the prior probability of a variant being associated with either GWAS ($p_1$) or gene expression ($p_2$). The prior probability ($p_{12}$) of a variant to be associated with both GWAS and gene expression was set at $1 \times 10^{-5}$. An eGene was declared to have colocalization when the posterior probability of colocalization of a GWAS variant and an eQTL linked to the eGene (PP4) was greater than 0.75.

## Fine mapping DICE eQTLs

We employed our custom fine-mapping pipeline (https://github.com/ay-lab/finemap) based on the package FINEMAP[20] with default settings to perform statistical fine-mapping of DICE eQTLs of various CD4 T cell subsets and their LD SNPs. We used the reference and alternate alleles, MAF, $P$ value, and beta values of individual SNPs. Fine-mapping was applied on individual genes and all the SNPs having association statistics with that gene. The package LDstore[116] was used to compute the LD statistics. FINEMAP was employed by allowing a maximum of ten causal SNPs (--n-causal-snps 10) and executed both stepwise conditioning (--cond) and shotgun stochastic search (--sss). An eQTL was defined as a fine-mapped variant if it overlapped with fine-mapping outputs from either stepwise conditioning or shotgun stochastic search approaches.

## Derivation of Multi-loop-iQTLs

We defined a multi-loop-iQTL as an iQTL satisfying all of the following conditions:

1. The SNP is associated with more than one HiChIP loops.
2. All these loops exhibit similar trend (and direction) of HiChIP contact counts for different genotypes, and
3. Genotype-dependent variation of the HiChIP contact counts combined for all the loops should be statistically significant.

To check whether a iQTL $X$ is a multi-loop-iQTL, we obtained the ratio values of observed and expected contact counts (from FitHiChIP statistics) from all its associated HiChIP loops and from all the input donors. We then computed the association between these ratios of contact counts with the genotypes of $X$ using linear regression (applying the *lm()* function of R). Associations with $p$-values < 1e-5 were considered significant and corresponding iQTLs were labeled as the multi-loop-iQTLs.

## Derivation of Connectivity-iQTLs

We defined a connectivity-iQTL as a iQTL if it exhibits significant association between its genotype and the contact counts of a group of HiChIP contacts (significant by FitHiChIP in at least one input sample) in the same direction, with respect to a broad region or locus $R$. To define such a testable region $R$ for a given iQTL $X$ associated with a HiChIP loop $L$, we adapted the connected component labeling algorithm (see[89] and also our previous work[64]) for identifying the set of HiChIP loops $S$ which are significant by FitHiChIP in at least one input sample and are spatially adjacent to the loop $L$. By spatial adjacency, we mean that the interacting bins of these loops are proximal. Formally, as the SNP $X$ is a iQTL of the loop $L$, $X$ exhibits genotype-dependent variation of HiChIP contact counts for $L$. We first initialized $S$ with the loop $L$. If the interacting 5kb bins of the loop $L$ are denoted by $b_1$ and $b_2$ (where, without loss of generality, $b_1 < b_2$), using all the loops (and their constituent bins) in $S$, we defined the following quantities: $r_1^s = \min(b_1 \forall L \in S)$; $r_1^e = \max(b_1 \forall L \in S)$; $r_2^s = \min(b_2 \forall L \in S)$; $r_2^e = \max(b_2 \forall L \in S)$, where min and max denote the conventional minimum and maximum operations. We defined the interacting segments **A** and **B** as **A** = [$r_1^s$, $r_1^e$] and **B** = [$r_2^s$, $r_2^e$]. Then we checked if there is any HiChIP loop $L'$ (between the interacting bins $b_1'$ and $b_2'$, $b_1' < b_2'$) not in $S$ such that the bin $b_1'$ lies within the interval [$r_1^s$ - $t$, $r_1^e$ + $t$] and the bin $b_2'$ lies within interval [$r_2^s$ - $t$, $r_2^e$ + $t$], where $t$ is a distance slack used to define the spatial proximity of a candidate loop to the set $S$. We used $t$ = 10kb, thus effectively provided a slack of two bins (since we used 5kb resolution for HiChIP loops). This procedure iteratively adds new loops $L'$ to the set $S$ based on the spatial proximity. Once the loop $L'$ is added to $S$, the quantities $r_1^s$, $r_1^e$, $r_2^s$, $r_2^e$ (and corresponding interacting segments **A** and **B**) are updated. The procedure converges when no new loop $L'$ can be added to $S$. For the final set of loops $S$ (and corresponding values of $r_1^s$, $r_1^e$, $r_2^s$, $r_2^e$), one of the following two possibilities occur:

1. $r_1^e < r_2^s$, i.e., the interacting segments **A** and **B** are disjoint. Here we define $S$ by the complete set of HiChIP loops (significant by FitHiChIP in at least one input sample) such that one ends of these loops belong to the segment **A** while the other ends belong to the segment **B**.
2. $r_1^e$ >= $r_2^s$, i.e., the interacting segments **A** and **B** actually overlap. So, here we consider $S$ as the set of loops whose endpoints belong to the complete span [$r_1^s$, $r_2^e$] (i.e., union of **A** and **B**).

Once the set of loops $S$ is constructed, we defined an iQTL $X$ as a *connectivity-iQTL* associated with the loops in $S$ provided the following conditions were satisfied: 1) ratios of observed and expected HiChIP contact counts for all the HiChIP contacts in $S$ followed similar trends of genotype-dependent variation, and their combined trend was also statistically significant, and 2) HiChIP contact matrices formed by using all the contacts between the regions **A** and **B** (when $r_1^e < r_2^s$) or within the region spanned by **A** and **B** (when $r_1^e >= r_2^s$) showed a substantial difference between genotypes (i.e., reference homozygous, heterozygous, and alternate homozygous) with respect to the iQTL $X$. Formally, we first derived the ratios of observed and expected contact counts (obtained from FitHiChIP statistics) for all the loops $L \in S$. These ratio values were then fitted against the genotypes of individual donors (with respect to the iQTL $X$) by linear regression (we used the R function *lm()*). For statistically significant regression outputs (*p*-value < $1e^{-5}$ and Bonferroni-corrected *p*-value < 0.005), we used the complete set of HiChIP contacts (contact count > 0, may or may not be significant by FitHiChIP in any of the input samples) within the regions **A** and **B** to define an aggregated contact map. Specifically, if the regions **A** and **B** were distinct ($r_1^e < r_2^s$), the set of contacts had their first interacting bins at **A** and the other interacting bins at **B**. On the other hand, if the regions **A** and **B** overlapped ($r_1^e >= r_2^s$), we used all the contacts within the span $[r_1^s, r_2^e]$. These HiChIP contacts were used to define three aggregated contact maps $M_{ref}$, $M_{het}$, $M_{alt}$ with respect to three different genotypes (reference homozygous, heterozygous and alternate homozygous) with respect to the iQTL $X$. Rows and columns of these matrices corresponded to individual 5kb bins. Entries in the contact matrices were the mean values of the ratios of observed and expected HiChIP contact counts (obtained from FitHiChIP) for the corresponding donors having the specific genotype. We then defined the quantities $y_{ref}$, $y_{het}$, $y_{alt}$ as the 95th-percentiles of the $\log_2$-transformed respective contact maps. The iQTL $X$ was defined as the connectivity iQTL with respect to the loops in $S$ and the intervals **A** and **B** if $y_{ref}$, $y_{het}$, $y_{alt}$ were monotonic (increasing or decreasing) with a step of $>= Thr$ (where $Thr$ is a static threshold, defined 0.1 in this study), indicating that the aggregated contacts either steadily increased or decreased according to the genotypes (reference homozygous -> heterozygous -> alternate homozygous).

### Assessment of iQTL loci using deep learning-based 3D chromatin conformation methods

To validate the iQTLs or, in general, the regulatory regions containing iQTL loci, in terms of their impact on the 3D chromatin conformation, we used the recently published deep learning method C. Origami[86] which predicts 3D chromatin contact maps using DNA sequences and 1D epigenomic annotations (CTCF ChIP-seq, ATAC-seq or DNase-seq), by applying a combination of convolutional neural network (CNN) and transformer architectures. We generated a trained model using the routine *corigami-train* (with default parameters) on the following datasets: 1) hg38 fasta sequences, 2) CD4 T cell Hi-C data[59] from the merged 54 Hi-C libraries (available in http://bartzabel.ls.manchester.ac.uk/orozcolab/SNP2Mechanism/hic/merged/), 3) CD4 T cell ATAC-seq bigwig file (downloaded from https://www.encodeproject.org/files/ENCFF991WSF/@@download/ENCFF991WSF.bigWig), 4) CTCF fold change bigwig file (downloaded from https://www.encodeproject.org/files/ENCFF443YUJ/@@download/ENCFF443YUJ.bigWig). The preprocessing routine provided in C. Origami was used to convert the.hic file into.npz files compatible to the train routine. The file *centrotelo.bed* for hg38 was downloaded from the C. Origami GitHub repository. As the reference epigenomic datasets are from hg38 assembly, we had to convert the iQTL and connectivity-QTL (and corresponding loci) hg19 coordinates to hg38 using the UCSC

hgLiftOver (https://genome.ucsc.edu/cgi-bin/hgLiftOver) routine. We then used in silico genetic screening routine *corigami-screen* from C. Origami to understand which regions of perturbation lead to greatest impact of the 3D chromatin prediction. The *perturb-width* parameter, defined as the size of the deletion (in bp) was kept as 1000 (default settings) while the *step-size* parameter, defined as the distance (in bp) between two successive deletions, was also fixed as 1000 (default settings). As the deletion size is about 1kb, the module cannot accurately pinpoint the contribution of one particular SNP, but rather provides an estimate regarding the contribution of a particular loci (of a few kb sizes at least) for 3D chromatin organization. In view of this, we provided the loci containing the connectivity-QTLs (reported in Supplementary Data 10) and the set of iQTL loci reported in Figs. 2–4, Supplementary Fig. 5. The output of this routine is a bedgraph file, whose 4th column corresponds to the impact score (higher score means higher perturbation impact). These bedgraph tracks were converted back to hg19 coordinates using the hgLiftOver routine, and plotted as an additional track for these example loci (iQTLs and connectivity-QTLs).

### Using Enhancer to gene (E2G) links from the activity-by-contact (ABC) model

We downloaded the enhancer-to-gene (E2G) links for the activated naïve CD4 T cell type from the ENCODE accession number ENCSR165CCO, as provided in the study[66]. Specifically, we considered the file ENCFF820YRW.bed containing the E2G links filtered by the ABC score thresholds provided in refs. 55,66 and used UCSC hgLiftOver (https://genome.ucsc.edu/cgi-bin/hgLiftOver) routine to convert the enhancer intervals from hg38 coordinates to hg19, compute their overlap with iQTL SNPs, and plot as a track in WashU browser.

### Statistics and reproducibility

Statistical analysis and test details are mentioned in the figure legends. We used R and Microsoft Excel to conduct the statistical analysis. No statistical method was used to predetermine sample size. No data were excluded from the analyses. The experiments were not randomized. The Investigators were not blinded to allocation during experiments and outcome assessment.

### Reporting summary

Further information on research design is available in the Nature Portfolio Reporting Summary linked to this article.

## Data availability

Sequencing data: HiChIP sequencing data has been uploaded in dbGAP (https://www.ncbi.nlm.nih.gov/projects/gap/cgi-bin/molecular.cgi?study_id=phs001703.v5.p1) and the corresponding HiChIP samples are listed in Supplementary Data 12. ChIP-seq peaks and HiChIP loops: ChIP-seq peaks used in this study are shared in Supplementary Data 2. Complete set of FitHiChIP loops from all the donors are provided at the Zenodo repository https://doi.org/10.5281/zenodo.13127086 (file *Complete_FitHiChIP_Loops_iQTL_Input.xlsx*). Complete set of iQTLs and connectivity-QTLs derived from this work are provided in Supplementary Data 4 and 10, respectively. WashU epigenome browser tracks: To visualize the shared tracks, user needs to open the WashU epigenome browser[99–101] (https://epigenomegateway.wustl.edu/browser/), and then load the session ID **c730ec60-4de0-11ef-8802-7f6b1b69f09b** containing the iQTLs, associated HiChIP loops, aggregate ChIP-seq peaks, FitHiChIP loops for individual donors, and the reference DICE nCD4 eQTLs. Web browser: We also created a web browser https://ay-lab-tools.lji.org/iQTL/ listing all the derived iQTLs, connectivity-QTLs, corresponding looping information, genotype and allele-specific trend plots, and the WashU browser tracks for individual SNP-loop pairs. Source data are provided with this paper.

## Code availability

All the developed software and scripts are available through a GitHub repository for this project (https://github.com/ay-lab/iQTL). The scripts for fine mapping GWAS and eQTL are hosted at https://github.com/ay-lab/finemap. Script for colocalization between eQTL and GWAS summary statistics is provided at https://github.com/ay-lab/Colocalization. The script for stratified LD Score Regression (SNP specific) is provided at https://github.com/ay-lab/S_LDSC_SNP. We have also uploaded the source code at the Zenodo repository https://doi.org/10.5281/zenodo.13127086 (folder: *SourceCode_GitHub*).

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

## Acknowledgements

We thank the La Jolla Institute (LJI) Flow Cytometry Core for assisting with cell sorting; the LJI's Clinical Studies Core for organizing sample collection; LJI's NGS core and Dr. Gregory Seumois for sequencing. We thank Dr. Benjamin Schmiedel for sample processing, cell sorting and ChIP-seq data, and Dr. Vivek Chandra for performing HiChIP experiments and contributing to the initial conception of the project under the supervision of Dr. Pandurangan Vijayanand (LJI). Additionally, we thank Dr. Chi-Hua Chen for valuable comments and edits on the manuscript. We also thank the members of Ay and Vijayanand labs for their valuable support. This work was funded by NIH grants R35-GM128938 (F.A.) and R24-AI108564 (MPI – F.A.). Utilized equipment was supported by the NIH grants no. S10-RR027366 (BD FACSAria II) and no. S10-OD016262 (Illumina HiSeq 2500).

## Author contributions

F.A. and S.B. conceived the work. S.B developed the computational algorithms and performed bioinformatic analysis under the supervision of F.A. S.B and F.A. wrote the manuscript. All authors have read and approved the contents of the manuscript.

## Competing interests

The authors declare no competing interests.
