## [Peer Review File · Nature Communications]

Identifying genetic variants associated with chromatin looping and genome functionReviewers' Comments:

Reviewer #1:

Remarks to the Author:

The work by Bhattacharyya and Ay builds on a previously published work by the same group (Chandra et al., 2021 in the list of references, although in the NatGen page is indicated as 2020) to describe what the authors call “interaction QTL (IQTL)”. These are genetic variants associated to altered HiChIP contacts and expression. These IQTLs could be considered, thus, structural QTL that associate changes in interaction between parts of the genome with expression level of interacting genes. Overall, this is an interesting approach that appears to be well conducted.

However, there are many aspects that limit its scope. First, the work is difficult to read and precludes the plausible reader to fully grasp the significance of the results. Additionally, at least to this reviewer, many of the sentences in the introduction are not clear as written. Second, to the exception of one dataset of HiChIP in naïve CD4 T cells, the rest of the data has already been generated and published by the same authors in previous works. Thus, this work shall be considered as re-analysis of existing data, which by itself is not a disadvantage. However, this reviewer could neither see significant novelty in the methods used. Third, the design of the approach seems a self-circular to this reviewer. By using HiChIP of H3K27ac it is clear that the authors are using a small sub-set of all loops in the genome that are more likely to be involved in QTL as H3K27ac is a proxy for a marker of active enhancers. Fourth, the authors present a large number of “overlapping” analysis that aim at demonstrating that the discovered IQTLs are functionally relevant. However, are the findings (i.e., % of overlaps with various datasets) significant? That has never been demonstrated. One would need to find out if such overlaps are significant using a contingency table or a similar statistical approach. For example, how many H3K27ac peaks that are NOT IQTL overlap with the many other factors studied? Is this more than expected compared to peaks that are in IQTLs? In other words, a % of an overlap, with no null model to compare with, is not very informative. As most of the results rely in such analysis, this would need to be fully addressed to properly assess whether the findings are significant. Finally, for the work to be novel and significant, this reviewer considers that the work would need to be done with Hi-C capturing all interactions in the genome and then re-classifying them instead of focusing from the start on H3K27ac interacting loops. Obviously, and as indicated by the authors, this is not a simple nor cheap experiment to be executed.

Reviewer #2:

None

Reviewer #3:

Remarks to the Author:

Bhattacharyya and Ay generated enhancer interaction data in naïve CD4 T cells using HiChIP in 30 healthy donors and demonstrated the link between genotypes and long-range interactions in these individuals. They reported that there is a substantial overlap between IQTL (defined as genotype-

dependent variation of HiChIP contacts and eQTLs (gene expression) or histone QTLs. In cases where IQTL in naïve CD4 T cells did not link to changes in gene expression, authors searched for other T cell subsets and found the link in gene expression in memory CD4 T cells for a subset of these loops. Additionally, they defined connectivity-QTL for larger genomic regions and reported that concordant genotype-dependent changes in chromatin contacts over a broad genomic region such as the RNASET2 region.

Overall, this is an important study and the large-scale profiling of enhancer-enhancer interaction map is very valuable for the community. Authors nicely took advantage of publicly available resources such OpenTarget etc to corroborate their findings. Considering the importance of T cell activation and the enrichment of SNPs associated with immune mediated diseases within enhancers of activated CD4 T cells, the choice of naïve CD4 T cells as a cell type to perform profiling is surprising. Having said that, these data demonstrate a resource for the community. I have two comments:

- 1) Considering that eQTL is typically defined by a fixed distance from genes, I am curious if modifying this distance can have an impact in the overlap between IQTLs and eQTLs in naïve T cells.
- 2) Considering the importance of the HiChIP data for the community, I recommend the authors to generate a webtool, sharing not only the browser views of their data but also their IQTLs.

Reviewer #4:

Remarks to the Author:

Bhattacharyya & Ay present a datasets comprised of naïve CD4+ T cell H3K27Ac HiChIP data for 30 healthy human donors. Premise of the paper is to use this relatively high number of individuals to identify interaction QTLs or IQTLs – SNP genotypes linked to quantitative differences in chromatin loop strength. They use these IQTLs to prioritize GWAS SNPs, yielding a partially different set from previous QTL prioritization effects (e.g. eQTL). Although I am not an expert in the underlying statistical methods, the methodology looks very solid and is coming from leading experts in the field. Main novelty and interest of this work is the generation of an IQTL resource (and associated methodology), which will be useful for the field.

Comments:

1. Ext Data Fig1: The authors define 600k loops, but over half (54%) are unique to individual donors. In a population of resting naïve T cells from healthy donors one expects to find fairly homogenous epigenomic landscapes, so I'm surprised by this level of donor-specific interactions. One explanation is that these interactions are false-positive calls representing weak and poorly reproducible signals that just reach the significance threshold. What statistical test underlies loop calling? Have the authors tried to call loops more stringently? Does this quickly lower the number of donor-specific loops (a sign of poorly reproducible calls)? Are donor-specific loops indeed weak and more variable loops?

2. Line 265-266: asthma is not an autoimmune disease, please correct.

3. It took me some time to figure out which SNPs are exactly labeled as associated with a HiChIP loop, it seems those within a 15kb window around both 5kb loop anchors? It would be helpful to show this is a small diagram in Fig.1.

4. Fig.2G is difficult to understand. Right now, it seems like the altered nucleotide is at the very non-critical edge of the motif. I would be very reluctant to suggest impact on binding from such a minor motif change - if my interpretation is correct, I suggest to remove this data. Panel H in contrast does show a disruption of a major motif nucleotide, which makes for a much more plausible scenario of TF binding disruption.

5. Fig.6 is referred to in the text prior to Fig.5, please resolve.

6. Line 544. I really appreciate this discussion on the potential mix of 1D/3D signals in HiChIP – something that is consistently ignored by part of the field. Overall, an excellent discussion paragraph.

Reviewer #5:

Remarks to the Author:

Sourya Bhattacharyya and Ferhat Ay generated HiChIP datasets on naïve CD4 T cells from 30 donors using an H3K27ac antibody. They identified interaction QTLs, indicating genotype-dependent and/or allele-specific variations associated with the intensity of chromatin contacts. Some of these QTLs were linked to eQTLs and hQTLs in naïve CD4 T cells or more committed CD4+ T cells.

The work is novel, featuring a solid computational methodology, and I really appreciate their effort to develop an interesting project with potential clinical impact. Importantly, it underscores the significance of chromatin contacts in QTL mapping, revealing innovative classes of connectivity-QTLs. However, I have several major and minor concerns:

Major points

1. Approximately 46% of loops are unique to one out of the 30 donors. I am aware of the limitation in reproducibility of the 3C-based methods, which is inherent of the methodology. For this reason, it is important to demonstrate data quality. Therefore, the comparison using PCA, as well as a correlation analysis, of these datasets with HiChIP data of other cell types will be helpful. For instance, the authors could use their H3K27ac HiChIP datasets from 5 major immune cell subsets (Chandra et al., 2021). These analyses should demonstrate higher similarities between nCD4 HiChIP datasets rather than between HiChIP datasets generated using different cell types. Finally, generation and analysis of two biological replicates from few of these donors (i.e., 2 independent HiChIP libraries from nCD4+ T cells isolated from the same donor, preferentially isolate in two

independent blood extractions if possible) will be required to assess the effect of experimental noise on the method's reproducibility.

2. It is not clear for me why the authors considered bi-allelic SNPs within +/-1 bin of each loop anchor (15kb region per anchor) instead of the SNPs on the 5kb anchor if the used HiChIP datasets have high resolution, complexity and quality. They should explain the underlying reason and demonstrate that this approach is not just a manner to pick up higher number of DNA loops associated with bi-allelic SNPs. If not, the +/-1 bin criteria is not correct under my point of view.

3. How many of the IQTLs SNPs overlap enhancers? It is worth to investigate it on top of the TF motive to provide mechanistic understanding. Blueprint and other consortiums have released genome-wide histone modification data to identify this type of distal regulatory elements in nCD4+ T cells to study this overlap.

How many of ~266k reference H3K27ac peaks overlap gene promoters? How many of ~600k statistically significant HiChIP engaged a gene promoter? I think that this information can provide a more comprehensive view of the results and can provide support of the transcriptional regulatory role of these bi-allelic SNPs engaged in differential looping.

4. The project is purely descriptive and any of the findings is functional validated. I am aware that this type of validations is time-consuming and difficult in some cases, even more for primary cells. However, functional validation inducing genetic perturbations in a cell line (instead of a primary T cells) will significantly increase the impact of the project.

5. The authors suggest that the changes in HiChIP contact counts are linked with the H3K27ac levels (line 191). Since the detection of HiChIP loops depends on the presence of H3K27ac (i.e., if the histone modification is not present in at least one of the anchors the loop cannot be detected), how the authors are sure that the looping strength fully depend on the genotype instead of being an effect of the dynamic deposition of this histone modification? Demonstrating anticorrelation between H3K27ac level at anchors and looping strength will be useful to avoid this concern. Besides, the incorporation on fig 2a, fig 2c, fig3c, fig 4b and fig 6a of three tracks of normalized H3K27ac counts (one for each SNP genotype) will support that the differences in looping are not an artifact of the differential H3K27ac levels. Besides, a track displaying the 5Kb windows should be included to provide a better idea of data resolution and interaction range.

6. The authors claim that previous studies profiling cell-specific 3D chromatin conformation in primary cells lacked the sample size for identifying genotype-dependent trends of chromatin contacts or measuring allele-specific differences. They also claim that most studies overlook genetic variants associated with epigenetic features but not gene expression in the considered cell type. However, this claim may not be entirely accurate, as demonstrated by Watt et al (Nat Commun 2021; doi: 10.1038/s41467-021-22548-8). Watt et al used PChi-C to investigate the impact of genetic variation on PU.1 binding, revealing an enrichment of transcription factor QTLs and histone QTLs, particularly the inhibitory mark H3K27me3, through allele-specific analysis of heterozygous sites near gene promoters via DNA looping.

7. On line 343-344 they claim “iQTLs may find a refined subset of eQTLs where the effect on gene expression can be attributed to underlying changes in chromatin contacts”. However, I am not sure if they can strongly claim it in all cases. For instance, fig 2a shows SNPs that control the expression of ORMDL3 gene and the strength of a DNA loop that connect GSDMB and IKZF3 genes. I cannot see a clear connection between this loop and ORMDL3 gene expression. A clarification in this case would be appreciated

Minor points.

8. It could be very useful for the scientific community to include on Table 5 the information about the genes whose promoters are associated in each DNA loop.

9. Please, homogenize font size and format across figures. For instance, color legend of extended figure 2A cannot be appreciated (i.e., circles are too small) or figure 2F is difficult to read.

10. More informative titles for each result sections could improve the manuscript quality.

Reviewer #1

The work by Bhattacharyya and Ay builds on a previously published work by the same group (Chandra et al., 2021 in the list of references, although in the NatGen page is indicated as 2020) to describe what the authors call “interaction QTL (IQTL)”. These are genetic variants associated to altered HiChIP contacts and expression. These IQTLs could be considered, thus, structural QTL that associate changes in interaction between parts of the genome with expression level of interacting genes. Overall, this is an interesting approach that appears to be well conducted.

Response:

We thank the reviewer for summarizing this work. In terms of the year of previous work, we list here the full citation which shows Epub was 2020 but it is listed as Jan 2021 in its final form: Nat Genet. 2021 Jan;53(1):110-119. doi: 10.1038/s41588-020-00745-3. Epub 2020 Dec 21

However, there are many aspects that limits its scope. First, the work is difficult to read and precludes the plausible reader to fully grasp the significance of the results. Additionally, at least to this reviewer, many of the sentences in the introduction are not clear as written.

Response:

We thank the reviewer for this warning. We have now streamlined and extensively shortened the Introduction (by about 1.5 pages). We also went over the text and did the same for other sections where either the sentences were too long or the flow was disrupted by sentences that are not critical.

Second, to the exception of one dataset of HiChIP in naïve CD4 T cells, the rest of the data has already been generated and published by the same authors in previous works. Thus, this work shall be considered as re-analysis of existing data, which by itself is not a disadvantage. However, this reviewer could neither see significant novelty in the methods used.

Response:

We would like to clarify that, in addition to existing published data, for this manuscript we generated H3K27ac HiChIP data from 24 donors with over 6.4 billion paired-end sequencing reads total (now available through dbGAP https://www.ncbi.nlm.nih.gov/projects/gap/cgi-bin/molecular.cgi?study_id=phs001703.v5.p1). We believe this is a significant undertaking and is a highly valuable dataset for studying T cells and gene regulation. Also, the mapping of genetic variants that are associated with looping has not been done before using HiChIP data and the methods we develop to accomplish this are also novel, in that sense. Below we describe in more detail the novelty of the generated data and the mentioned approach:

Novelty in the problem statement:

In our previous work (Chandra et al. Nature Genetics 2021), the objective was to infer the putatively functional expression QTLs (eQTLs) using HiChIP contacts. We used eQTLs of five different immune cell types (CD4 Naive T cell, CD8 Naive T cell, Classical Monocyte, Naive B and Natural Killer), and also generated H3K27ac HiChIP datasets of 6 individuals for each of

these cell types. Using the HiChIP contacts, we classified a subset of eQTLs as putatively functional by virtue of their HiChIP contacts to the respective eGenes.

In the current work, we have proposed a novel class of QTLs (which may or may not be eQTLs) having genotype-dependent variation of HiChIP contacts between individuals or allele-specific variation of HiChIP reads. To establish the methodology and framework, we focused on naïve CD4+ T cells, and in addition to the previously published HiChIP data from 6 donors, we generated HiChIP data for another 24 donors, and used these 30 HiChIP datasets to model individual specific differences of chromatin contacts and the SNPs associated with such changes. Thus, the objectives, the datasets used and the methodologies developed in these two studies are quite different.

Novelty in the generated data:

We generated H3K27ac HiChIP data from **24** donors with over **6.4** billion paired-end sequencing reads total (now available through dbGAP https://www.ncbi.nlm.nih.gov/projects/gap/cgi-bin/molecular.cgi?study_id=phs001703.v5.p1, as mentioned in the data availability section). This new HiChIP data on its own is a substantial resource that has not been generated by any other group for any primary cell type to the best of our knowledge. No study so far employed or generated individual-specific HiChIP datasets to decode the role of genetic variants in altering chromatin interactions.

Novelty in the developed approach:

- As the objective of this study is to derive SNPs associated with individual-specific variation of HiChIP contacts, we have developed a novel computational approach to infer these SNPs. Our method is the first to comprehensively employ both genotype-dependent variation of chromatin contacts and allele-specific variation of HiChIP reads to identify SNPs associated with individual-specific variation of HiChIP contacts. Thus, we do not agree with the assessment that this work is simply “reanalysis of existing data”, this may likely be a misunderstanding.
- In addition to our work, two other new studies both in preprint as of 2023, *Shi et al.* (<https://www.medrxiv.org/content/10.1101/2023.07.19.23292550v1.full>) and *Ray-Jones et al.* (<https://www.biorxiv.org/content/10.1101/2023.08.04.551251v2>) also developed independent frameworks for identifying SNPs associated with the changes in chromatin contacts, using Hi-C and capture Hi-C datasets, respectively. Our paper was initially submitted in coordination with those two other papers.
- We have proposed a comprehensive categorization of iQTLs based on whether they are eQTLs of the naive CD4 T cell, eQTLs in other T cell subsets, or not. Such a characterization showed us that iQTLs in naive CD4 T cells can be eQTLs in other T cell subsets (Treg, Tfh etc.) indicating that the genotype or allele-specific variation of chromatin contacts in naive CD4 T cell are translated to gene expression level in the derived T cell subsets. This concept and characterization of such iQTLs is novel and not previously described neither in published work nor the two preprints listed above.
- Using HiChIP contacts, we also provided a novel concept of connectivity-QTLs where QTLs were shown to regulate not only one HiChIP contacts but a group of HiChIP contacts

spanning a broad genomic region. Using locus-specific examples (*RNASET2*), we showed the concordance between gene expression and the synchronous trends of multiple chromatin contacts over, thereby suggesting that concordant changes in chromatin contact can lead to changes in gene expression. Again, this concept has never been proposed before or investigated in neither previously published work nor the two preprints listed above.

Third, the design of the approach seems a self-circular to this reviewer. By using HiChIP of H3K27ac it is clear that the authors are using a small sub-set of all loops in the genome that are more likely to be involved in QTL as H3K27ac is a proxy for a marker of active enhancers.

Response:

We thank the reviewer for this comment and would like to clarify our motivation in using HiChIP. Our starting point is that H3K27ac (active enhancer and promoter mark) driven loops are among not only the most relevant loops for gene regulation but are also more likely to vary among different genotypes in accordance with gene expression. By enriching such loops, we are obtaining sufficient resolution to study regulatory elements with manageable sequencing depth, which enabled this study in the first place.

We, however, acknowledge that (and this has been mentioned in the Discussion section as well) the detected H3K27ac loop changes may relate to both 1D and 3D signals. Therefore, an iQTL effect can be modulated in a mixture of two possible ways: one being through H3K27ac levels and the other through loop strength. To understand whether the observed iQTL are accompanied by similar changes 1D histone modifications, we have performed multiple new analyses and are highlighting old and new results related to this issue:

1. We already presented our analysis in Extended Data Figs. 3B, 3E that not all iQTLs are H3K27ac hQTLs. Specifically, 72% iQTLs do not overlap such hQTLs and 62% do not have any LD with them ($LD < 0.8$). Therefore, for a majority of iQTL loops, the genotype-dependent variation is related to changes in 3D conformation.
2. To further assess the impact of iQTL associated loops in regulating 3D chromatin conformation, we utilized the aggregated CD4 T cell Hi-C contact map made available by Shi et. al. preprint (medRxiv 2023) and performed aggregate peak analysis (APA) on both iQTL associated loops and the complete set of background FitHiChIP loops. APA results showed that iQTL associated loops are supported strongly by Hi-C data similar to what we have shown with HiChIP data before (now added at Extended Data Fig. 2D).
3. We have now extensively analyzed the genotype-dependent and allele-specific variation of 1D H3K27ac ChIP-seq and HiChIP coverage for various iQTLs and connectivity QTLs, and incorporated them in Figs. 2B (*ORMDL3* locus), 2D (*CD247* locus), 4C (*RUNX3* locus), Extended data figs. 5D (*TRAF1* locus), 6C (*MICB* locus), 8B (*RNASET2* locus). In all of these examples, however, we did not find statistically significant genotype-dependent or allele-specific deviation of 1D ChIP-seq and HiChIP coverage, suggesting that the changes in chromatin looping due to iQTLs or connectivity-QTLs cannot be attributed to 1D changes and they happen at the 3D genome level.

Fourth, the authors present a large number of “overlapping” analysis that aim at demonstrating that the discovered IQTLs are functionally relevant. However, are the findings (i.e., % of overlaps with various datasets) significant? That has never been demonstrated. One would need to find out if such overlaps are significant using a contingency table or a similar statistical approach. For example, how many H3K27ac peaks that are NOT IQTL overlap with the many other factors studied? Is this more than expected compared to peaks that are in IQTLs? In other words, a % of an overlap, with no null model to compare with, is not very informative. As most of the results rely in such analysis, this would need to be fully addressed to properly assess whether the findings are significant.

Response:

We thank the reviewer for this comment. First, we would like to clarify that our overlap analysis of iQTLs with eQTLs (or hQTLs) was to categorize these QTLs further rather than testing whether such overlaps are statistically significant. Nevertheless, according to the suggestion, we have added a supplementary method section “**Significance of various overlap analysis**” (describing the details of various null models) and also **Supplementary Table 12** containing the details of overlap and contingency tables. We have also added the significance analysis for various overlap statistics in the main text, as summarized below:

1. Results Section A, paragraph 4 - we have modified the main text as: “*We observed that ~64% (5933 out of 9426) iQTL SNPs overlap with eQTLs or hQTLs (Fisher’s exact test p-value < 0.00001, OR = 1.43; see **Methods; Table 12A**)*”.
2. Results Section A, paragraph 4 - we compared the overlap of reference CD4 Naive eQTLs and hQTLs with respect to three different sets of anchors (5kb bins): 1) Loop anchors (5kb bins) for the iQTL associated HiChIP loops, 2) control set of FitHiChIP loop anchors such that these loops are not associated with any IQTLs, and 3) 5kb anchors randomly sampled from the genomic intervals. We also compared the overlap of reference CD4 Naive eQTLs and hQTLs with respect to two different sets of ChIP-seq peaks: 1) ChIP-seq peaks falling in the iQTL associated HiChIP loop anchors, 2) Rest of the ChIP-seq peaks. The significance analysis has now been updated in the main text as: “*Overall, iQTL-associated HiChIP loop anchors showed higher overlap with reference CD4 Naive DICE eQTLs or hQTLs compared to either FitHiChIP loop anchors not associated with IQTLs (OR: 1.39, Fisher’s exact test p-value < 0.00001) or random anchors (OR: 5.14, Fisher’s exact test p-value < 0.00001) (see **Methods, Table 12C**). The same was true when we compared between the ChIP-seq peaks in the iQTL associated HiChIP loop anchors versus the rest, in terms of their overlap with reference eQTLs or hQTLs (OR: 2.46, Fisher’s exact test p-value < 0.00001) (see **Methods, Table 12D**).*”
3. Results Section B, paragraph 1 - regarding the overlap of iQTLs and naive CD4 T eQTLs, we modified the main text as: “*We observed that 5381 iQTLs (~57%) are also eQTLs of nCD4 (Fisher’s exact test p-value < 0.00001, OR: 1.88, see **Methods; Table 12B**)*”
4. Results Section B, paragraph 1 - regarding the fraction of regulatory SNPs for the iQTLs which are also nCD4 eQTLs, we have now computed the statistical significance, and updated the main text as: “*Indeed, we observed that 1725 iQTLs (~32%) of this category fall within the regulatory regions defined by aggregate nCD4 H3K27ac ChIP-seq peaks,*

and ChIP-seq peaks for various CD4 T cell subsets and regulatory TFs provided in the ChIP-Atlas(Zou et al., 2022) database (see **Methods** and **Table 5**), and such a fraction of regulatory SNPs is substantially higher (OR: 3.16; Fisher's exact test p-value < 0.00001; **Table 12E**) than that of the reference DICE nCD4 eQTLs (~13% regulatory SNPs) (**Table 6**)."

5. Results Section B, paragraph 1 - we have now computed the statistical significance of the overlap of iQTLs (which are also nCD4 eQTLs) and nCD4 eQTLs with the TF binding motifs reported by FIMO, AME and ADASTRA. We updated the main text as: "*Considering iQTLs which are also nCD4 eQTLs, ~62% SNPs overlapped with TF binding sites derived by at least one of these approaches (compared to ~51% DICE Naïve CD4 eQTLs; OR: 1.58; Fisher's exact test p-value < 0.00001; **Table 12F**), out of which ~19% SNPs showed TF binding motif evidence from at least two approaches (compared to 1.3% DICE Naïve CD4 eQTLs; OR: 17.3; Fisher's exact test p-value < 0.00001; **Table 12F**) (**Extended data fig. 4A**).*"
6. Results Section B, paragraph 2 - we have already reported the Fisher's exact test p-values regarding the overlap of fine-mapped GWAS SNPs with the iQTLs (which are also Naive CD4 T cell eQTLs) compared to the full set of DICE Naive CD4 T cell eQTLs, with respect to various immune diseases.
7. Results Section C, paragraph 1 - we have reported the statistical significance regarding the overlap with regulatory regions between iQTLs (which are not naive CD4 T cell eQTLs but eQTLs in other CD4 T cell subsets) and complete DICE naive CD4 T eQTLs. We updated the main text as: "*In total, these 1143 iQTLs were associated with 482 distinct HiChIP loops. Around 34% (386) of these iQTLs overlapped with regulatory peaks (**Table 5**), a percentage comparable to iQTLs which are nCD4 eQTLs, and much higher (OR: 3.41; Fisher's exact test p-value < 0.00001; **Table 12E**) compared to the complete set of DICE nCD4 eQTLs. Around 60% of iQTLs in this category overlapped TF binding motifs supported by at least one of the techniques FIMO, AME or ADASTRA TF (OR 1.45, Fisher's exact test p-value < 0.00001, compared to the complete set of DICE nCD4 eQTLs, **Table 12F**) and ~20% had motif overlap supported by at least two methods (OR 18.5; Fisher's exact test p-value < 0.00001, compared to the complete set of DICE nCD4 eQTLs, **Table 12F**) (**Fig. 3A** and **Table 5**)".*
8. Results Section D, paragraph 1 - we now added the enrichment statistics for the iQTLs exclusive to any CD4 T cell eQTLs, and updated the main text as: "*In terms of overlap with regulatory regions (~30%) (OR: 2.87, Fisher's exact test p-value < 0.00001 with respect to the complete DICE nCD4 eQTLs, **Table 12E**) and TF binding motifs (FIMO analysis: 61% with 20% supported by two databases, odd ratios of 1.51 and 18.45, respectively, with respect to the complete DICE nCD4 eQTLs; **Table 12F**)".*

Finally, for the work to be novel and significant, this reviewer considers that the work would need to be done with Hi-C capturing all interactions in the genome and then re-classifying them instead of focusing from the start on H3K27ac interacting loops. Obviously, and as indicated by the authors, this is not a simple nor cheap experiment to be executed.

Response:

We acknowledge the comment from the reviewer but respectfully disagree with the point that the novelty and significance of this work, or any work, can be one-to-one associated with the use of a specific experimental technique. Among the three papers submitted in coordination with this work, our paper used HiChIP, Orozco lab used Hi-C (<https://www.medrxiv.org/content/10.1101/2023.07.19.23292550v1.full>) and Spivakov lab used Capture Hi-C (<https://www.biorxiv.org/content/10.1101/2023.08.04.551251v2>), all with interesting and novel findings, in our opinion. We do clearly explain our motivation in focusing on a subset of interactions (and enriching for them) and also explicitly discuss the advantages and limitations of our work. We also mention earlier work that attempted mapping loop/interactions QTLs using Hi-C data, however, lacked the sequencing depth, sufficient resolution and/or sample numbers required to identify QTLs impacting loops directly and deferred back to more general properties of 3D organization such as insulation, TADs or compartments. We started this work 5 years ago and HiChIP has been the technology that enabled such studies at that time. We hope the reviewer appreciates the difficulty associated with such large-scale studies using human samples.

Reviewer #3

Bhattacharyya and Ay generated enhancer interaction data in naïve CD4 T cells using HiChIP in 30 healthy donors and demonstrated the link between genotypes and long-range interactions in these individuals. They reported that there is a substantial overlap between IQTL (defined as genotype-dependent variation of HiChIP contacts and eQTLs (gene expression) or histone QTLs. In cases where IQTL in naïve CD4 T cells did not link to changes in gene expression, authors searched for other T cell subsets and found the link in gene expression in memory CD4 T cells for a subset of these loops. Additionally, they defined connectivity-QTL for larger genomic regions and reported that concordant genotype-dependent changes in chromatin contacts over a broad genomic region such as the RNASET2 region. Overall, this is an important study and the large-scale profiling of enhancer-enhancer interaction map is very valuable for the community. Authors nicely took advantage of publicly available resources such OpenTarget etc to corroborate their findings.

Response:

We thank the reviewer for their thorough and helpful assessment, and for highlighting the novelty and the importance of our work.

Considering the importance of T cell activation and the enrichment of SNPs associated with immune mediated diseases within enhancers of activated CD4 T cells, the choice of naïve CD4 T cells as a cell type to perform profiling is surprising. Having said that, these data demonstrate a resource for the community.

Response:

We agree that studying activated T cells (or memory subsets) would also have been interesting. However, our justification for studying naive cells was in twofold:

1. Naive CD4 T cells provide a more homogeneous population and avoids potential confounders that can be introduced in case of in vitro activation, which are important factors that are difficult to explicitly account for.
2. By studying the naive CD4 T cells without any activation or differentiation/polarization, we aimed to identify iQTLs that are potentially the underlying eQTLs for such activation and differentiation conditions as performing an iQTL study for each such condition would have been prohibitive. We achieved this goal in finding iQTLs that become eQTLs in specific T helper subsets (12% of iQTLs).

I have two comments:

1) Considering that eQTL is typically defined by a fixed distance from genes, I am curious if modifying this distance can have an impact in the overlap between iQTLs and eQTLs in naïve T cells.

Response:

We thank the reviewer for this valuable comment. Although eQTLs in literature are almost exclusively defined as having distance < 1 Mb from the TSS of the respective genes, in our earlier work (*Chandra et al. Nature Genetics 2021*) we derived ultra-long range eQTLs beyond 1 Mb from TSS and are interacting with the promoter in the same set of naïve CD4 T cells. Upon reviewers' suggestion, when we overlapped our iQTLs with these ultra-long range eQTLs, we only obtained 2 additional eQTLs that are in LD with an iQTL and no eQTLs that overlap with an iQTL. Therefore, the impact of including beyond 1Mb eQTLs will be negligible on the overlap analysis. This finding is also consistent with the fact that only 30 out of 2292 iQTL loops we report have a genomic distance greater than 1Mb. These results could be explained by either most variants having impact on more local interactions or more likely by the diminishing power of detecting meaningful variation in ultra-long-range loops, which have much lower contact counts than shorter range loops.

2) Considering the importance of the HiChIP data for the community, I recommend the authors to generate a webtool, sharing not only the browser views of their data but also their iQTLs.

Response:

We thank the reviewer for this great suggestion. Given the thousands of iQTLs we found, separating donors for each QTL into genotype groups and presenting this data has been challenging. We, however, have now cataloged the list of iQTLs and also the list of connectivity QTLs in the webserver: <https://ay-lab-tools.lji.org/iQTL/>. The server contains separate links to the pages displaying iQTLs and connectivity-QTLs, and also contains detailed documentation for individual summary statistics.

For the iQTLs, in addition to the SNP information, we also list the associated HiChIP loops, genotype and allele-specific trend visualization plots, and the WashU epigenome browser links for individual entries.

Similarly, for the connectivity-QTLs, we list the associated HiChIP loops and the broad locus information for each case, and provide the respective WashU epigenome browser links and observed / expected contact count trends for different genotypes.

We have also added search functionality in these pages, so that the user can search individual entries by gene, SNP or loop coordinates. We hope that this browser view would be useful to the community. We thank the reviewer again for their constructive suggestion on making this data more easily accessible.

Reviewer #4

Bhattacharyya & Ay present a datasets comprised of naive CD4+ T cell H3K27Ac HiChIP data for 30 healthy human donors. Premise of the paper is to use this relatively high number of individuals to identify interaction QTLs or IQTLs – SNP genotypes linked to quantitative differences in chromatin loop strength. They use these IQTLs to prioritize GWAS SNPs, yielding a partially different set from previous QTL prioritization effects (e.g. eQTL). Although I am not an expert in the underlying statistical methods, the methodology looks very solid and is coming from leading experts in the field. Main novelty and interest of this work is the generation of an IQTL resource (and associated methodology), which will be useful for the field.

Response:

We thank the reviewer for summarizing this work and appreciating its novelty.

Comments:

1. Ext Data Fig1: The authors define 600k loops, but over half (54%) are unique to individual donors. In a population of resting naive T cells from healthy donors one expects to find fairly homogenous epigenomic landscapes, so I'm surprised by this level of donor-specific interactions. One explanation is that these interactions are false-positive calls representing weak and poorly reproducible signals that just reach the significance threshold. What statistical test underlies loop calling? Have the authors tried to call loops more stringently? Does this quickly lower the number of donor-specific loops (a sign of poorly reproducible calls)? Are donor-specific loops indeed weak and more variable loops?

Response:

We thank the reviewer for the comment and acknowledge with one minor correction - 46% of ~600K FitHiChIP loops (union from all the samples, significant in at least one sample) are unique to individual donors (**Extended Data Fig. 1A**). To call the significant chromatin loops, we used our previously published method FitHiChIP (Bhattacharyya et al. Nature Communications 2019) with loose (L) background, where the complete set of peak-to-all contacts (one end overlapping with peak regions) were used as the background. To check whether a stringent loop calling alters the fraction of chromatin interactions exclusive to only one donor, we executed FitHiChIP with the stringent (S) background which employs the peak-to-peak contacts (i.e., contacts where both anchors overlap with the regulatory regions) therefore returns a lower number of significant loops (due to more stringent background model). We obtained ~393K loops (~65% loops of the loose

background) which are significant in at least one input sample. However, we observed exactly the same percentage (46%) of FitHiChIP(S) loops significant in only one donor (**Response Figure 1**) as compared to FitHiChIP(L).

Response Figure 1: Overlap of FitHiChIP(S) significant loops among different donors employed in the iQTL study.

We then focussed on the FitHiChIP(L) loops employed in the iQTL study, and compared the ~46% FitHiChIP(L) loops exclusive to one donor to the rest of the FitHiChIP(L) loops. We observed that although these two sets of loops exhibit similar distance distributions (**Response Figure 2**), mean contact counts of these ~46% loops (exclusive to one donor) are lower than those of the rest of the loops for most of the distance bins (**Response Figure 3**), suggesting that the loops exclusive to individual donors are in general weaker.

Response Figure 2: Comparison of distance distribution of FitHiChIP(L) loops significant exclusively in one donor vs the rest of the FitHiChIP(L) loops significant in at least two input donors.

Response Figure 3: Comparison of median contact counts of FitHiChIP(L) loops significant exclusively in one donor vs the rest of the FitHiChIP(L) loops significant in at least two input donors.

We, however, note that most of these exclusive loops are not associated with iQTLs. Out of the 2292 HiChIP loops associated with the iQTLs, only 375 (~16%) loops are significant exclusively in one donor (**Extended Data Fig. 2A**). Further, loops associated with iQTLs also have higher median contact counts and are significant in higher numbers of donors compared to the complete set of FitHiChIP loops used as the background for iQTL discovery (**Extended Data Fig. 2B**). These 2292 loops also exhibit higher enrichment of the HiChIP background (higher APA scores) compared to the rest of the FitHiChIP loops (**Extended Data Fig. 2C**). These results indicate that our iQTL analysis prioritizes mainly the loops that are reproducible across multiple donors and are stronger than the average HiChIP loop.

2. Line 265-266: asthma is not an autoimmune disease, please correct.

Response:

We thank the reviewer for pointing out this error. We have replaced the phrase “many autoimmune diseases, most notably in asthma” with “in asthma as well as a number of autoimmune diseases” at Section B, paragraph 3.

3. It took me some time to figure out which SNPs are exactly labeled as associated with a HiChIP loop, it seems those within a 15kb window around both 5kb loop anchors? It would be helpful to show this is a small diagram in Fig.1.

Response:

We thank the reviewer for this great suggestion. We have added a schematic diagram **Fig. 1(C)** showing that for each HiChIP loop between two 5kb loop anchors, SNPs within the 15kb region surrounding each anchor (+/- 1 bin from each anchor) are tested for association with the HiChIP loop. We have also referred to this figure at Section A, paragraph 2 of the revised manuscript.

4. Fig.2G is difficult to understand. Right now, it seems like the altered nucleotide is at the very non-critical edge of the motif. I would be very reluctant to suggest impact on binding from such a minor motif change - if my interpretation is correct, I suggest to remove this data. Panel H in contrast does show a disruption of a major motif nucleotide, which makes for a much more plausible scenario of TF binding disruption.

Response:

We thank the reviewer for this suggestion, and accordingly we have removed Fig. 2G and its reference at the last paragraph of Section B of the revised manuscript. We have also updated the reference of the previous Fig. 2H to Fig. 2G in the same paragraph. However, we would like to note that regions immediately flanking TF binding motifs (through their flexibility and structure) have been shown to have large effects on physical binding of such TFs (Yella et. al. Nucleic Acids Research 2018, PMC6294565). Since this is not central to our study, we decided to not discuss this further and removed the figure panel as suggested.

5. Fig.6 is referred to in the text prior to Fig.5, please resolve.

Response:

We thank the reviewer for pointing out this issue. We have inserted the following text referring Fig. 5 at section E, paragraph 1 of the revised manuscript: *“We thus enquired whether there exist iQTLs associated with not only one loop but rather with multiple HiChIP loops in a broad locus, and devised a method to infer such QTLs (Fig. 5)”*.

6. Line 544. I really appreciate this discussion on the potential mix of 1D/3D signals in HiChIP – something that is consistently ignored by part of the field. Overall, an excellent discussion paragraph.

Response:

We thank the reviewer for acknowledging the discussion section and our assessment of 1D/3D HiChIP signals. We completely agree that this is an important point that has to be kept in mind for interpretation of results from HiChIP and similar enrichment-based conformation capture assays.

Reviewer #5

Sourya Bhattacharyya and Ferhat Ay generated HiChIP datasets on naïve CD4 T cells from 30 donors using an H3K27ac antibody. They identified interaction QTLs, indicating genotype-dependent and/or allele-specific variations associated with the intensity of chromatin contacts. Some of these QTLs were linked to eQTLs and hQTLs in naïve CD4 T cells or more committed CD4+ T cells.

The work is novel, featuring a solid computational methodology, and I really appreciate their effort to develop an interesting project with potential clinical impact. Importantly, it underscores the significance of chromatin contacts in QTL mapping, revealing innovative classes of connectivity-QTLs.

Response:

We thank the reviewer for summarizing this work and appreciating its novelty.

Major points

1. Approximately 46% of loops are unique to one out of the 30 donors. I am aware of the limitation in reproducibility of the 3C-based methods, which is inherent of the methodology. For this reason, it is important to demonstrate data quality. Therefore, the comparison using PCA, as well as a correlation analysis, of these datasets with HiChIP data of other cell types will be helpful. For instance, the authors could use their H3K27ac HiChIP datasets from 5 major immune cell subsets (Chandra et al., 2021). These analyses should demonstrate higher similarities between nCD4 HiChIP datasets rather than between HiChIP datasets generated using different cell types. Finally, generation and analysis of two biological replicates from few of these donors (i.e., 2 independent HiChIP libraries from nCD4+ T cells isolated from the same donor, preferentially isolate in two independent blood extractions if possible) will be required to assess the effect of experimental noise on the method's reproducibility.

Response:

We thank the reviewer for this suggestion. We have previously shown that applying PCA on HiChIP contact counts of all 30 samples did not produce any outliers (**Extended Data Fig. 1B**). We note that these 30 samples include our previously published 6 naive CD4+ T HiChIP samples (Chandra et al. *Nature Genetics* 2021) and the newly generated 24 HiChIP samples.

We have now performed PCA using all 30 naive CD4 iQTL samples and all the HiChIP samples from CD8 Naive, Monocyte, Naive B and Natural Killer cell types (n=6 each) provided in our earlier study (Chandra et al. *Nature Genetics* 2021), thus utilizing 54 samples in total. We used the same set of HiChIP loops as used in our earlier work (Fig. 1B of Chandra et al. *Nature Genetics* 2021) and computed the contact counts and statistical significance values of the iQTL samples for these loops. PCA with respect to the ratio of observed and expected contact counts shows that the naive CD4 samples (used in the current iQTL study) are separated from the samples of other cell types (**Response Figure 4**).

Response Figure 4: PCA using the ratio of observed and expected contact counts with respect to the loops used in our earlier study (Fig. 1B of Chandra et al. Nature Genetics 2021) for the current iQTL samples and previously published HiChIP samples of 4 other cell types.

Also, in our previous study (Chandra et al. Nature Genetics 2021), we reported the reproducibility of HiChIP contact maps between replicates from the same donor. With respect to the **Extended Data Fig. 1B** of the manuscript Chandra et al. Nature Genetics 2021, the results were stated as: “Reproducibility analysis of HiChIP contact counts among replicates of same donor for naive CD4+ T cells ($n = 3$) and classical monocytes ($n = 3$) showed a correlation (0.81–0.87; Extended Data Fig. 1b) suggesting reproducibility of chromatin interaction”.

2. It is not clear for me why the authors considered bi-allelic SNPs within +/-1 bin of each loop anchor (15kb region per anchor) instead of the SNPs on the 5kb anchor if the used HiChIP datasets have high resolution, complexity and quality. They should explain the underlying reason and demonstrate that this approach is not just a manner to pick up higher number of DNA loops associated with bi-allelic SNPs. If not, the +/-1 bin criteria is not correct under my point of view.

Response:

We thank the reviewer for this comment. At the Section A, end of paragraph 2, we have inserted the following text: “We note that out of 2292 HiChIP loops associated with the iQTL SNPs, the majority (1684 loops, ~73%) contain the iQTLs in their interacting anchors (5kb) as opposed to

the flanking bins that cover twice the size (+/-5kb)". As this result suggests, the main reason behind our approach was not simply to pick up a higher number of HiChIP loops or iQTLs.

Various existing studies like hichipper (*Lareau et al. Nature Methods 2018*), hichip-peaks (*Shi et al. Bioinformatics 2020*) suggested that HiChIP reads tend to come from regions proximal to the restriction sites and some may not be contained within the specific loop anchors, particularly for high resolution (5kb) data. Further, the work FIND (*Djekidel et. al. Genome Research 2018*) pointed out that due to the polymeric nature of the chromatin fiber, for high resolution (such as 5kb) contact matrix, establishment of a chromatin loop that brings two interacting loci (i, j) into spatial proximity would also influence the spatial distance between their adjacent loci ($i - 1$ and $i + 1$ for the locus i ; $j - 1$ and $j + 1$ for the locus j). In view of this, we also tested the association of SNPs in the adjacent 5kb bins ($i - 1, i + 1$ and $j - 1, j + 1$) with respect to the HiChIP contact between the bin pairs (i, j). Considering all of these prior studies and due to using high resolution (5kb) HiChIP contact maps for the iQTL analysis, we opted for including the SNPs within +/- 1 bin of the HiChIP loop anchors, to test their associations with the corresponding loops.

3. How many of the iQTLs SNPs overlap enhancers? It is worth to investigate it on top of the TF motive to provide mechanistic understanding. Blueprint and other consortiums have released genome-wide histone modification data to identify this type of distal regulatory elements in nCD4+ T cells to study this overlap.

Response:

We thank the reviewer for this comment and state that in the result sections B, C and D, we have already mentioned the number of iQTLs falling within the regulatory regions defined by aggregate nCD4 H3K27ac ChIP-seq peaks, and ChIP-seq peaks for various CD4 T cell subsets and regulatory TFs provided in the ChIP-Atlas database. We have also included per-SNP statistics regarding their overlap with the regulatory regions, in the Supplementary Table 5. Here we are re-stating the numbers. Specifically, for iQTLs overlapping with nCD4 eQTLs, 1725 iQTLs (~32%) fall within such regulatory regions (result section B). For iQTLs not eQTLs in nCD4 but eQTLs for other CD4 T cell subsets, ~34% of these iQTLs overlapped with regulatory regions (result section C). For iQTLs which are not eQTLs in any CD4 T cells, ~30% iQTLs overlap with regulatory regions (result section D).

According to the reviewer's suggestions, we have now also included the non-redundant hg19 ChIP-seq peaks from the ReMap2022 database (<https://remap2022.univ-amu.fr/>) and specifically considered the peaks marked for CD4 or T cells. We found that ~14% (1287 out of 9426) iQTL SNPs fall within these peaks, including 371 new SNPs which did not overlap with the previously mentioned DICE nCD4 H3K27ac ChIP-seq peaks or peaks in the ChIP-Atlas database. We have included SNP-wise overlap statistics with respect to these ReMap ChIP-seq peaks in the Supplementary Table 5 (column AL).

We have also separately computed the exact overlap of iQTLs with the H3K27ac ChIP-seq peaks (~266k from aggregated ChIP-seq data across donors) used in our study (which are also used

for FitHiChIP loop calling), and also the hg19 BLUEPRINT histone peaks downloaded from the IHEC data portal (<https://epigenomesportal.ca/ihec>) for various histone marks such as H3K27ac, H3K27me3, H3K36me3, H3K4me1, H3K4me3 and H3K9me3 (n=2 donors each). Specifically, we downloaded two samples with accession IDs MS000401 and MS000501. For these reference sets of peaks, we computed the fraction of iQTLs overlapping peaks, and also profiled the median distance between iQTLs with their nearest peaks. The following table summarizes these quantities suggesting one third of the iQTLs overlap H3K27ac peaks, 26% overlap H3K4me1 and 28% overlap H3K4me3 for the MS000501 sample with higher number of ChIP-seq peaks. The percent overlap of iQTLs with our H3K27ac ChIP-seq peaks is at 46% reflecting the larger number of peaks called compared to IHEC data.

Study	Histone	Total number of reference peaks	Number and fraction of iQTLs overlapping loop anchors with ChIP-seq peak	Median distance between iQTLs and nearest peaks (bp)
Current	Aggregated H3K27ac ChIP-seq	266426	4372 (0.46)	1131
IHEC - MS000401	H3K27ac	47121	2894 (0.3)	3530
	H3K27me3	48922	138 (0.01)	115976
	H3K36me3	44685	920 (0.1)	14747
	H3K4me1	16964	885 (0.1)	30665
	H3K4me3	17121	1440 (0.15)	12839
	H3K9me3	613	25 (0.003)	3.76*10 ⁶
IHEC - MS000501	H3K27ac	87905	3345 (0.35)	2007
	H3K27me3	13366	64 (0.006)	194167
	H3K36me3	42354	972 (0.1)	13408
	H3K4me1	61672	2440 (0.26)	4035
	H3K4me3	37582	2601 (0.28)	4321

	H3K9me3	196	18 (0.002)	5.95*10 ⁶
--	---------	-----	------------	----------------------

4. How many of ~266k reference H3K27ac peaks overlap gene promoters? How many of ~600k statistically significant HiChIP engaged a gene promoter? I think that this information can provide a more comprehensive view of the results and can provide support of the transcriptional regulatory role of these bi-allelic SNPs engaged in differential looping.

Response:

We thank the reviewer for this suggestion. As per the comment, we analyzed the overlap of the reference ChIP-seq peaks, iQTLs and HiChIP loops, with the reference set of TSS. We considered a 500 bp window either side of TSS for overlap analysis.

- 65,353 ChIP-seq peaks (~25%) overlap with the TSS of 26,715 promoters.
- 54,084 ChIP-seq peaks (~20%) overlap with the TSS of 16,461 protein-coding promoters.

We then annotated the ~600K HiChIP loops (specifically their anchors) according to the following criterion:

- If the loop anchor overlaps with the reference protein coding TSS +/- 500 bp, the anchor is labeled as promoter (P).
- If a non-promoter anchor overlaps with the reference ChIP-seq peaks, it is annotated as enhancer (E).
- Otherwise, the anchor is labeled as other (O).

With respect to this criterion, we observed the following classification of loops:

- 250,400 loops (~42%) are P-P loops.
- 180,821 loops (~30%) are P-E or E-P loops.
- 80,824 loops (~13%) are E-E loops.
- 60,692 loops (~10%) are P-O or O-P loops.
- Remaining loops (~5%) are E-O or O-E loops.

We also applied the same analysis on the iQTL associated 2292 HiChIP loops, and found the following categorization:

- 908 loops (~40%) are P-P loops.
- 693 loops (~30%) are P-E or E-P loops.
 - 1135 iQTLs (~12%) fall within the Promoters of these P-E loops
 - 1222 iQTLs (~13%) fall within the Enhancers of these P-E loops
- 312 loops (~14%) are E-E loops.
- 257 loops (~11%) are P-O or O-P loops.
- Remaining loops (~5%) are E-O or O-E loops.

Finally, we also checked the overlap of iQTL SNPs with the promoters:

- Considering the complete set of reference TSSs, we observed 2622 (~28%) SNPs overlap fall within 500 bp of the reference set of promoters which is consistent with the overlap we report for H3K4me3 peaks from IHEC data.

5. The project is purely descriptive and any of the findings is functional validated. I am aware that this type of validations is time-consuming and difficult in some cases, even more for primary cells. However, functional validation inducing genetic perturbations in a cell line (instead of a primary T cells) will significantly increase the impact of the project.

Response:

We thank the reviewer for this comment and also for understanding the difficulties in conducting these functional validation studies. To predict the regulatory impact of iQTLs, Connectivity-QTLs and corresponding loci, we employed the deep-learning method C. Origami (Tan et al. Nature Biotechnology 2023; PMID: 36624151) which uses DNA sequence and epigenomic data (ATAC-seq and CTCF ChIP log fold change tracks) to predict 3D contact maps. The trained model can then be used to perform *in-silico* genetic screening, by stepwise deletion of genomic segments (of a fixed resolution) and scoring the relative changes on the 3D chromatin conformation.

We used CD4 T cell Hi-C data provided by Shi et. al. preprint (medRxiv 2023) to first generate a trained model, using the default routines and configuration options of C. Origami. Using this model, we performed *in-silico* genetic screening with a resolution of 1kb and a step size of 1kb, for all the example iQTL and connectivity-QTL loci provided in the main and supplementary figures. The idea is, for each iteration, 1000 bp sequence information would be deleted (perturbation) and a new chromatin contact map would be estimated using the perturbed sequence as an input to the trained model. The difference between the original and the predicted Hi-C contact maps would serve as the *impact score*, where a higher impact score indicates that corresponding 1000 bp genomic region has a high impact in 3D chromatin organization. The section *Assessment of iQTL loci using deep learning-based 3D chromatin conformation methods* in the supplementary methods describes running the C. Origami package and applying the screening process in detail.

As the perturbation screening is carried out at a resolution of 1000 bp, assessing the contribution of individual iQTL SNPs was not possible. However, we could measure the impact of individual loci containing the iQTLs and connectivity-QTLs. For example, we screened the *RNASET2* locus (140kb region in chr6) containing the connectivity QTL rs2247325 (Fig. 6A) and observed that the region having the highest impact score overlaps with this QTL, and also harbors CD4 T cell ATAC-seq peaks. In the *NUB1* locus (Extended Data Fig. 9A) the iQTLs rs11980001 and rs6464142 and the surrounding 5kb anchors fall within the regions having the highest impact scores. In the *MICB* locus, iQTLs such as rs4947321 which are not eQTLs in naive CD4 T cell but eQTLs in other CD4 T cell subsets, are also proximal to the region with the highest impact score (Fig. 3C).

We, however, note two important limitations of this analysis:

1. The screening is performed with respect to the Hi-C data while the iQTLs are derived from the H3K27ac HiChIP data, so the iQTLs (and underlying loci) are not always expected to capture significant change in genome-wide chromatin conformation measured by Hi-C. In

addition, the high-depth Hi-C data used for training was from total CD4 T cells versus our HiChIP data was from naive CD4 T cells.

2. The deep learning technique and the prediction model has its limitations, given that the model is not trained on actual perturbation experiments as such data is extremely limited.

We also turned to another recent approach, that is the activity-by-contact (ABC) model (Fulco et al. Nature Genetics 2019 - PMID: 31784727; Nasser et al. Nature 2021 - PMID: 33828297; Gschwind et al. bioRxiv 2023), which has become popular in predicting enhancer-to-gene (E2G) links across different cell types. In view of this, to assess the regulatory potential of the derived iQTLs, we checked their overlap with the E2G links of the activated naive CD4 T cell type (discussed in the supplementary methods section *Using Enhancer to gene (E2G) links from the activity-by-contact (ABC) model*). We added the following texts in the manuscript, corresponding to the overlap between different categories of iQTLs and these enhancers:

- Section A, paragraph 2 - overall overlap between iQTLs and ABC enhancers: *“To further characterize iQTL SNPs, we also assessed their overlap with enhancers predicted by activity-by-contact (ABC) model (Gschwind et al., 2023). Around 40% (3765) of iQTLs were within 5kb (14% or 1279 with exact overlap) of enhancer regions of the enhancer-to-gene (E2G) links predicted by ABC model for the activated CD4 Naïve cells (see **Methods**)”.*
- Section B, paragraph 1 - considering the overlap of these ABC enhancers with the iQTLs which are eQTLs in naive CD4: *“Out of the 5381 iQTLs in this category, 2163 (~40%) were within 5kb of ABC enhancers (725 or ~13.5% with exact overlap) (Gschwind et al., 2023)”.*
- Section C, paragraph 1 - considering the overlap of these ABC enhancers with the iQTLs which are eQTLs in other CD4 T cells: *“Overall, 474 iQTLs (~41%) of this category overlap are within 5kb of ABC enhancers (198 or ~17% with exact overlap) of activated CD4 Naïve T cells”.*

In addition, we also added the tracks of ABC enhancers in Fig. 2C (*CD247* locus), Fig. 3C (*MICB* locus), Fig. 6A (*RNASET2* locus), Extended data fig. 9A (*NUB1* locus) and showed the overlap of iQTLs and connectivity-QTLs on the ABC enhancers linked to the corresponding genes. All these results support the notion that iQTLs and corresponding loci have impact on 3D chromatin organization and gene regulation.

6. The authors suggest that the changes in HiChIP contact counts are linked with the H3K27ac levels (line 191). Since the detection of HiChIP loops depends on the presence of H3K27ac (i.e., if the histone modification is not present in at least one of the anchors the loop cannot be detected), how the authors are sure that the looping strength fully depend on the genotype instead of being an effect of the dynamic deposition of this histone modification? Demonstrating anticorrelation between H3K27ac level at anchors and looping strength will be useful to avoid this concern. Besides, the incorporation on fig 2a, fig 2c, fig3c, fig 4b and fig 6a of three tracks of normalized H3K27ac counts (one for each SNP genotype) will support that the differences in looping are not an artifact of the differential H3K27ac levels. Besides, a track displaying the 5Kb windows should be included to provide a better idea of data resolution and interaction range.

Response:

We thank the reviewer for this great suggestion. We acknowledge that (and this has been mentioned in the Discussion section as well) the detected H3K27ac loop changes may relate to both 1D and 3D signals. Therefore, an iQTL effect can be modulated in a mixture of two possible ways: one being through H3K27ac levels and the other through loop strength. We already presented our analysis in Extended Data Fig. 3(D) that not all iQTLs are H3K72ac hQTLs (72% of iQTLs do not overlap such hQTLs and 62% do not have any LD with them ($LD < 0.8$)), therefore, for a majority of iQTL loops, the genotype-dependent variation is related to changes in 3D conformation. We have now also utilized the aggregated CD4 T cell Hi-C contact map made available by Shi et. al. preprint (medRxiv 2023) and performed aggregate peak analysis (APA) on both iQTL associated loops and the complete set of background FitHiChIP loops, which showed that iQTL associated loops are supported strongly by Hi-C data, even higher than the background HiChIP loops (Extended Data Fig. 2D).

Regarding the 5kb windows tracks, top of every WashU browser tracks provided in the respective figures indicate the genomic regions and coordinates.

According to the reviewer's suggestion, we have now extensively analyzed the genotype-dependent and allele-specific variation of 1D H3K27ac ChIP-seq and HiChIP coverage for various iQTLs and connectivity QTLs, and incorporated them in Figs. 2B (*ORMDL3* locus), 2D (*CD247* locus), 4C (*RUNX3* locus), Extended data figs. 5D (*TRAF1* locus), 6C (*MICB* locus), 8B (*RNASET2* locus). In all of these examples, however, we did not find statistically significant genotype-dependent or allele-specific deviation of 1D ChIP-seq and HiChIP coverage, suggesting that the changes in chromatin looping due to iQTLs or connectivity-QTLs cannot be attributed to 1D changes and happen at the 3D genome level.

7. The authors claim that previous studies profiling cell-specific 3D chromatin conformation in primary cells lacked the sample size for identifying genotype-dependent trends of chromatin contacts or measuring allele-specific differences. They also claim that most studies overlook genetic variants associated with epigenetic features but not gene expression in the considered cell type. However, this claim may not be entirely accurate, as demonstrated by Watt et al (Nat Commun 2021; doi: 10.1038/s41467-021-22548-8). Watt et al used PCHi-C to investigate the impact of genetic variation on PU.1 binding, revealing an enrichment of transcription factor QTLs and histone QTLs, particularly the inhibitory mark H3K27me3, through allele-specific analysis of heterozygous sites near gene promoters via DNA looping.

Response:

We thank the reviewer for pointing out this reference paper. We agree our second sentence "epigenetic features" was too broad and we rewrote this paragraph now as follows by highlighting the difference between 1D coverage-associated SNPs versus loop-associated SNPs (this work): "Published work that profile cell-specific 3D chromatin conformation (Chandra et al., 2021; Javierre et al., 2016; Jung et al., 2019; Watt et al., 2021) in primary cells has done it on a limited number of donors ($n=2-6$) making it difficult to identify de novo associations between genotype

and chromatin contacts. Among these Watt et al. (Watt et al., 2021) on primary neutrophils mapping transcription factor binding QTLs for PU.1 utilized allele-specific bias of promoter capture Hi-C (PCHi-C) data but did that on 1D coverage (all PCHi-C reads) of the heterozygous SNPs rather than individual chromatin loops involving the SNP region as an anchor. Similarly, our previous work on 6 donors also considered allele-specific variation of 1D coverage from HiChIP for a set of examples (Chandra et al., 2021).”

This Watt et al. reference also brought to our attention the RNASET2 locus that is also studied there for neutrophils and monocytes in relation to a PU.1 tfQTL that was in perfect LD with lead GWAS SNP for IBD and Crohn's disease (rs2149092). Upon checking our data, we also found that rs2149092 is an iQTL in our data from naive CD4 T cells. However, since it is not a connectivity QTL and the discussion in this section is about connectivity QTLs, we did not include this information in the text. We are happy to do so if deemed necessary.

8. On line 343-344 they claim “iQTLs may find a refined subset of eQTLs where the effect on gene expression can be attributed to underlying changes in chromatin contacts”. However, I am not sure if they can strongly claim it in all cases. For instance, fig 2a shows SNPs that control the expression of ORMDL3 gene and the strength of a DNA loop that connect GSDMB and IKZF3 genes. I cannot see a clear connection between this loop and ORMDL3 gene expression. A clarification in this case would be appreciated

Response:

We thank the reviewer for this comment. We agree that this may not always be the case and hence used we used “iQTLs may find ...” to highlight this. We now further added “for some loci, iQTLs may find ...” to further soften the statement.

With respect to the *ORMDL3* locus the difficulty of explaining the relation to expression comes from multiple factors. First, the iQTLs rs2305479 and rs7216389 associated with the 40kb HiChIP loop between the genes *IKZF3* and *GSDMB* show significant association with HiChIP contacts (Figs. 2A-B) and gene expression (eQTL for *ORMDL3* - Extended data fig. 5A). They are also in LD with two SNPs rs4065275 (chr17:38080865) and rs12936231 (chr17:38029120), which we previously have shown to overlap with CTCF binding sites, and switch the CTCF binding from the gene *ZBP2* to *ORMDL3* intronic region in the risk haplotype (G for rs4065275 and C for rs12936231) compared to non-risk alleles (A for rs4065275 and G for rs12936231) for asthma. Therefore, although the promoter of *ORMDL3* is about 15kb away from this current loop endpoint the SNP rs4065275 overlapping *ORMDL3* is in LD with the iQTL SNPs of the 40kb loop and would have been an iQTL itself if it was closer to the loop anchor. Second issue is that there is indeed a 60kb loop that connects *ORMDL3* promoter to *IKZF3* promoter and using RASQUAL default model, this loop was also found to be significantly associated with rs12936231, which would have made rs12936231 an iQTL SNP with more direct association to *ORMDL3* expression. However, due to our extensive additional filtering of iQTL associations to minimize false positives, this association gets filtered out later.

To clarify the above discussion in the text, to an extent, we have added in Section B, paragraph 3 of the main text: “we observed that the two SNPs we previously identified to overlap with CTCF binding sites in naïve CD4 T cells (Schmiedel et al., 2016), namely rs4065275 (chr17:38080865) and rs12936231 (chr17:38029120) were also in tight LD with iQTLs but not the lead eQTL (Extended data fig. 5B). These two SNPs were shown to switch the CTCF binding from the gene ZPBP2 to ORMDL3 intronic region in the risk haplotype (G for rs4065275 and C for rs12936231) compared to non-risk alleles (A for rs4065275 and G for rs12936231).”

Minor points.

9. It could be very useful for the scientific community to include on Table 5 the information about the genes whose promoters are associated in each DNA loop.

Response:

We thank the reviewer for this comment. We have now included the genes overlapping individual loop anchors in Table 5 (column V). We have also now cataloged the list of iQTLs and also the list of connectivity QTLs in the webserver: <https://ay-lab-tools.lji.org/iQTL/>. The server contains separate links to the pages displaying iQTLs and connectivity-QTLs and lists corresponding genes

For the iQTLs, in addition to the SNP information, we also list the associated HiChIP loops, genotype and allele-specific trend visualization plots, and the WashU epigenome browser links for individual entries.

Similarly, for the connectivity-QTLs, we list the associated HiChIP loops and the broad locus information for each case, and provide the respective WashU epigenome browser links and observed / expected contact count trends for different genotypes.

We have also added search functionality in these pages, so that the user can search individual entries by gene, SNP or loop coordinates. We hope that this browser view would be useful to the community.

10. Please, homogenize font size and format across figures. For instance, color legend of extended figure 2A cannot be appreciated (i.e., circles are too small) or figure 2F is difficult to read.

Response:

We thank the reviewer for the suggestion. We have updated Fig. 2F and extended figure 2A accordingly. We have also revised other figures considering this comment, specifically Figs. 2D, Extended data figs. 3D, 3E, 3G.

11. More informative titles for each result sections could improve the manuscript quality.

Response:

We thank the reviewer for this comment. We have now updated the titles for each of the result sections, and also included section numbers (A,B,C) for more clarity. The updated section titles are:

- A. Interaction QTLs (iQTLs): a novel class of QTLs associated with 3D chromatin interactions
- B. iQTLs overlapping with eQTLs in naïve CD4 T cells are enriched for fine-mapped GWAS SNPs and TF binding motifs
- C. Naïve CD4 T cell iQTLs pinpoint cell-specific eQTLs in other CD4 T cell subsets, revealing potential regulation of gene expression by chromatin interactions
- D. iQTLs not eQTLs in any CD4 T cell subset are enriched for fine-mapped GWAS SNPs
- E. iQTLs may be associated with concordant changes in chromatin contacts over a broad genomic locus

Reviewers' Comments:

Reviewer #1:

Remarks to the Author:

I appreciate the authors have addressed to certain extend my original major concerns.

Reviewer #3:

Remarks to the Author:

Authors addressed my previous concerns.

Reviewer #4:

Remarks to the Author:

The authors have adequately addressed my questions and I now support publication.

Reviewer #5:

Remarks to the Author:

All my questions have been addressed. I have not further suggestions. Congratulations for the work.